# Choices are More Important than Effort: LLMs Enable Efficient Multi-Agent Exploration

## Abstract

With expansive state-action spaces, efficient multi-agent exploration remains a longstanding challenge in reinforcement learning. Although pursuing novelty, diversity, or uncertainty attracts increasing attention, redundant efforts brought by exploration without proper guidance choices poses a practical issue for the community. This paper introduces a systematic approach, termed LEMAE, choosing to channel informative task-relevant guidance from a knowledgeable **L**arge Language Model (LLM) for **E**fficient **M**ulti-**A**gent **E**xploration. Specifically, we ground linguistic knowledge from LLM into symbolic key states, that are critical for task fulfillment, in a discriminative manner at low LLM inference costs. To unleash the power of key states, we design Subspace-based Hindsight Intrinsic Reward (SHIR) to guide agents toward key states by increasing reward density. Additionally, we build the Key State Memory Tree (KSMT) to track transitions between key states in a specific task for organized exploration. Benefiting from diminishing redundant explorations, LEMAE outperforms existing SOTA approaches on the challenging benchmarks (e.g., SMAC and MPE) by a large margin, achieving a 10x acceleration in certain scenarios. Our code is available at https://anonymous.4open.science/r/LEMAE.

## 1 Introduction

Exploration stands as a fundamental issue in reinforcement learning (RL) (Du et al., 2023; Liu et al., 2023). Researchers have developed several exploration strategies directed by novelty, diversity, or uncertainty (Linke et al., 2020; Burda et al., 2018b; Pathak et al., 2017), mainly in single-agent reinforcement learning. However, these methods may induce task-irrelevant redundant exploration, especially in complex environments (Du et al., 2023). In the realm of Multi-Agent Reinforcement Learning (MARL), the need to mitigate exploration redundancy becomes even more urgent due to the challenges like exponential expansion of the state-action spaces. Widespread real-world applications, including MOBA games (Qu et al., 2023), social science (Jaques et al., 2019), and multi-vehicle control (Xu et al., 2018), further underscore the growing need for efficient multi-agent exploration.

This work identifies *task-relevant guidance* as an important consideration in enhancing exploration efficiency. Incorporating priors in exploration mechanism design, such as complex reward structures, typically requires expert knowledge and substantial human efforts (Liu et al., 2023; Abbeel & Ng, 2004). Hopefully, recent advances have witnessed the remarkable reasoning and planning capabilities of Large Language Models (Touvron et al., 2023; Achiam et al., 2023), providing a plausible choice to facilitate efficient exploration through LLM's effortless prior provision. However, it is non-trivial to effectively comprise linguistic LLM priors into symbolically represented RL tasks (Peng et al., 2023; Carta et al., 2023), and the investigation of practical ways to avoid nuisances caused by such an expression discrepancy is of critical importance.

In response to the above issue, we propose LEMAE, a novel framework to enable efficient multi-agent exploration with LLM. The framework primarily consists of two components: (i) *key states localization with LLM* and (ii) *key state-guided exploration*. The first component automatically localizes key states essential for task completion, through LLM discrimination, thereby injecting LLM priors into the RL process. Specifically, the discriminator function induced by LLM works to discriminate key states from rollout trajectories, avoiding the overburden of LLM inference costs. The second component harnesses the localized key states as meaningful guidance to achieve efficient

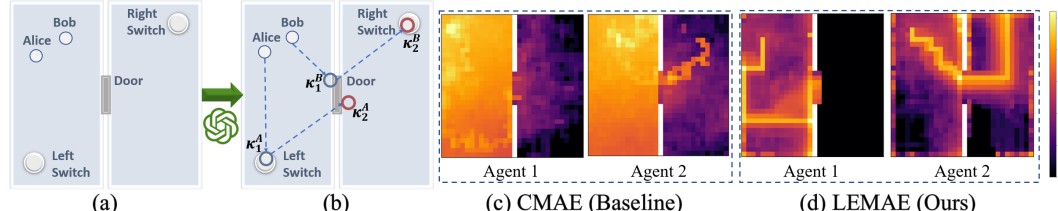

|     |     |     |     |
|-----|-----|-----|-----|
| (a) | (b) | (c) CMAE (Baseline) | (d) LEMAE (Ours) |

Figure 1: (a) The map of the task *Pass*. Two agents are initially positioned in the left room, requiring cooperation to explore the rooms, uncover the **hidden** switches, and move to the right room. (b) The key states ($\kappa_1$ and $\kappa_2$) generated by LLM for the task *Pass*, where the superscripts $A, B$ of $\kappa_i$ denote two agents Alice and Bob. (c) Visitation Map (log scale) of SOTA baseline method CMAE. (d) Visitation Map (log scale) of our method LEMAE. Our method exhibits a significant reduction in redundant exploration. Furthermore, an organic division of labor among agents emerges.

exploration. In implementation, we treat key states as explicit training signals, devising Subspace-based Hindsight Intrinsic Reward (SHIR) to refine rewards for achieving these key states. To facilitate organized exploration with memory, Key States Memory Tree (KSMT) is further constructed to track key state transitions, constraining the state space in exploration and enhancing guidance in SHIR. As illustrated in Fig. 1, our design empowers LEMAE with a significant performance advantage through notably reducing redundant exploration.

Our **main contributions** are summarized as follows:

1. We build a bridge between LLM and RL to facilitate efficient multi-agent exploration by developing a systematic approach dubbed LEMAE.

2. We devise a computationally efficient inference strategy channeling task-specific information from LLM to distinguish key states critical for task fulfillment as subgoals for targeted exploration.

3. We introduce a Key State Memory Tree to organize exploration according to historic key state transitions and devise the Subspace-based Hindsight Intrinsic Reward, encouraging agents' guidance.

We conduct extensive experiments on typical multi-agent exploration benchmarks. LEMAE (i) consistently outperforms the state-of-the-art (SOTA) baselines with **10x acceleration** in certain scenarios, (ii) achieves performance comparable to the baseline trained with human-designed dense rewards in **sparse reward** scenarios, and (iii) exhibits potential to generalize to tasks previously unseen by LLM or involving non-symbolic states. These observations validate the effectiveness of our design in reducing redundant exploration and improving exploration efficiency, showing promise for real-world deployment in scenarios requiring efficient exploration.

## 2 PRELIMINARY

The environments considered in this work are characterized as a decentralized partially observable Markov decision process (Dec-POMDP) (Oliehoek et al., 2016) with $n$ agents, which can be defined as a tuple $G = \langle S, A, I, P, r, Z, O, n, \gamma \rangle$, where $s \in S$ is the global state, $A$ is the action space for each agent, and $\gamma \in [0, 1)$ is the discount factor. At time step $t$, each agent $i \in I \equiv \{1, ..., n\}$ has its local observations $o^i \in O$ drawn from the observation function $Z(s, i) : S \times I \to O$ and chooses an action $a^i \in A$ by its policy $\pi^i(a^i|o^i) : O \to \Delta([0, 1]^{|A|})$, forming a joint action $\mathbf{a} \in \mathbf{A} \equiv A^n$. $T(s'|s, \mathbf{a}) : S \times \mathbf{A} \times S \to [0, 1]$ is the environment's state transition distribution. All agents share a common reward function $r(s, \mathbf{a}) : S \times \mathbf{A} \to \mathbb{R}$. The agents' joint policy $\boldsymbol{\pi} := \prod_{i=1}^{n} \pi^i$ induces a joint *action-value function*: $Q^{\boldsymbol{\pi}}(s, \mathbf{a}) = \mathbb{E}[R|s, \mathbf{a}]$, where $R = \sum_{t=0}^{\infty} \gamma^t r_t$ is the expected discounted return. The goal of MARL is to find the optimal joint policy $\boldsymbol{\pi}^*$ such that $Q^{\boldsymbol{\pi}^*}(s, \mathbf{a}) \geq Q^{\boldsymbol{\pi}}(s, \mathbf{a})$, $\forall \boldsymbol{\pi}$ and $(s, \mathbf{a}) \in S \times \mathbf{A}$. Notably, we specifically focus on sparse reward tasks, i.e., $r_t = 1$ only when $s_{t+1} = s_{success}$, otherwise $r_t = 0$. We denote the symbol for the $i$-th key state by $\kappa_i$ together with its discriminator function $\mathcal{F}_i$.

## 3 RELATED WORKS

**LLM in Decision Making.**  Large Language Models have shown impressive capabilities across downstream tasks (Touvron et al., 2023; Radford et al., 2019; Brown et al., 2020). Recent advances indicate a growing trend of using LLM in decision-making problems (Wang et al., 2023b). A primary challenge is grounding LLM's linguistic knowledge into specific low-level control tasks typically represented in symbolic form (Peng et al., 2023; Carta et al., 2023), especially in RL. Creating linguistic twin tasks (Carta et al., 2023) is intuitive but require substantial manual workloads. Some works employ LLMs as high-level planners, e.g., coding with APIs (Liang et al., 2023), using human-annotated or LLM-summarized actions (Yao et al., 2022; Shinn et al., 2023; Lin et al., 2023; Zhu et al., 2023; Wang et al., 2023a). Despite significant progress, they rely on difficult-to-obtain low-level policies or APIs, limiting real-world applicability. Recently, LLMs have been integrated with RL to enhance low-level decision making (Cao et al., 2024). LLMs act as environmental information processors, reducing learning complexity (Paischer et al., 2022; 2024; Kim et al., 2024; Wang et al., 2024), but cannot directly facilitate efficient exploration. Some works utilize LLMs as goal selectors in goal-conditioned RL (Su & Zhang, 2023; Shukla et al., 2023), teacher policy (Zhou et al., 2023), or task sampler (Zhang et al., 2023b) but require predefined task pools, skills or subgoals. Other methods like LLM-based reward or policy design (Klissarov et al., 2023; Ma et al., 2023; Kwon et al., 2023; Song et al., 2023; Liu et al., 2024; Chen et al., 2024) rely on large annotated datasets, frequent LLM inferences, or are restricted to simple tasks with ample information. Fine-tuning (Carta et al., 2023; Shi et al., 2023) demands enormous data and resources. LLaMAC (Zhang et al., 2023a) employs multiple LLMs to balance exploration and exploitation but emphasizes step-wise decision-making via frequent LLM calls. ELLM (Du et al., 2023) enhances exploration using LLM but depends on predefined symbolic observation captioner and frequent LLM inferences. Its semantic similarity-based rewards may also struggle to generalize across scenarios. In contrast, LEMAE integrates linguistic LLM priors into symbolic states with minimal task information and LLM inference costs, achieved by localizing key states in rollout trajectories using LLM-generated discriminator functions.

**Efficient Multi-Agent Exploration.**  Exploration efficiency has long been a focal point in RL (Thrun, 1992; Cai et al., 2020; Seo et al., 2021; Mahajan et al., 2019; Jeon et al., 2022; Ecoffet et al., 2019). Typical exploration methods focus on random exploration (Mnih et al., 2013; Rashid et al., 2018) or heuristic indicators, such as diversity or novelty, to facilitate exhaustive exploration, particularly in single agent exploration (Linke et al., 2020; Burda et al., 2018b; Pathak et al., 2017; Burda et al., 2018a; Bellemare et al., 2016). Despite their success, they may induce notable redundant exploration due to a lack of task-relevant guidance (Du et al., 2023). The exponential expansion of the state-action spaces renders exhaustive exploration impractical in multi-agent settings. Consequently, efficient multi-agent exploration (MAE) becomes increasingly imperative and necessary (Jeon et al., 2022; Liu et al., 2021). MAE is also challenging due to the complex configurations, e.g., the entangled effect of multi-agent actions and intricate reward design (Liu et al., 2023; Qu et al., 2023; Xu et al., 2023). Given our emphasis on efficient exploration, we prioritize evaluation in multi-agent settings. Some MAE methods encourage influential behaviors during agent interactions (Liu et al., 2023; Jaques et al., 2019; Wang et al., 2019). Nevertheless, they may lead to unintended coalitions or require additional priors (Liu et al., 2023). Certain studies leverage subgoals to guide exploration (Jeon et al., 2022). However, due to challenges in integrating task-related information into subgoals, they either necessitate human expertise for subgoals design (Tang et al., 2018; Kulkarni et al., 2016) or struggle to identify useful subgoals (Jeon et al., 2022; Liu et al., 2021). A recent influential work, WToE (Dong et al., 2023), focuses on when to explore by identifying discrepancies between the actor policy and an environmental-change-detection policy, which does not employ intrinsic rewards as guidance. Distinguished from the above, this work underscores the significance of task-relevant guidance in exploration and utilizes the key state priors extracted from LLM to enable efficient multi-agent exploration.

## 4 METHOD

This section first induces the concept of key states as task-relevant guidance. Centering around the key states, we construct two components: (i) *key states localization with LLM* (Sec. 4.2) and (ii) *key state-guided exploration* (Sec. 4.3). The former directs LLM to generate discriminator functions for localizing key states in rollout trajectories, while the latter guides exploration with the introduced Subspace-based Hindsight Intrinsic Reward and Key States Memory Tree. Please refer to Fig. 2 and Algorithm 1 for details. Also, we provide a demonstration to clarify LEMAE's execution pipeline.

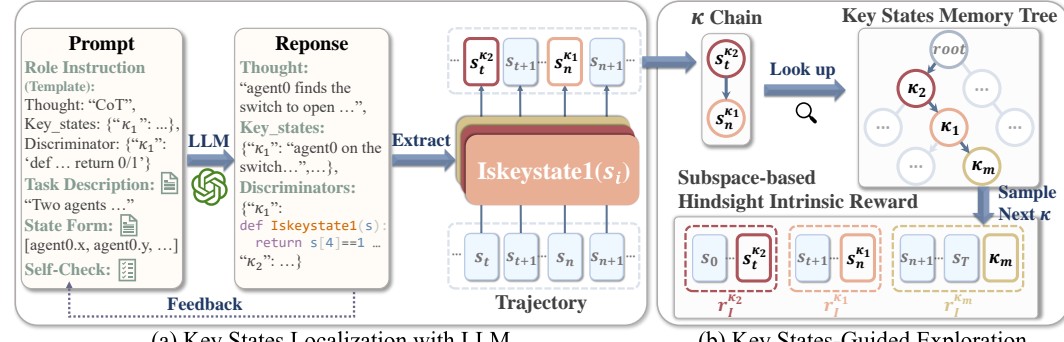

(a) Key States Localization with LLM     (b) Key States-Guided Exploration

Figure 2: Overview of the training process. (a) Key States Localization with LLM: We devise a set of prompts to guide LLM in localizing key states based on task-specific information. Refinements of the response are achieved through iterative self-checks by LLM. Subsequently, discriminator functions are derived from the final response to discriminate key states within trajectories. (b) Key States-Guided Exploration: Using the achieved key states chain within the processed trajectory, we look up KSMT to get the most probable next key states. By sampling from them as the subgoal for the concluding sub-trajectory, we integrate intrinsic rewards into the overall trajectory using SHIR.

## 4.1 Devil is in the Key States

Previous methods suffer from redundant exploration efforts in pursuing task-agnostic novelty (Du et al., 2023), potentially reducing training efficiency. This motivates us to integrate task-relevant information as a better guidance choice for efficient exploration. Nevertheless, practical proposals are limited in the field. This work identifies the **Key States** as the novel task-relevant prior, which corresponds to intermediate states with explicit semantics and expressions pertaining to the task. Meanwhile, Proposition 4.1 explicitly reflects the efficacy of incorporating them.

**Proposition 4.1.** *Consider the one-dimensional asymmetric random walk problem, where an agent starts at $x = 0$ and aims to reach $x = N \in \mathbb{N}^+, N > 1$. The initial policy is asymmetric and random with probabilities $p \in (0.5, 1)$ and $1 - p$ for right and left movements, respectively. Without prior knowledge, the expected first hitting time is $\mathbb{E}(T_{0 \to N}) = \frac{N}{2p-1}$. After introducing the task-relevant information that the agent must first reach key states $\kappa = 1, ..., N-1$ before reaching $x = N$, we can decrease the expected first hitting time by $\mathbb{E}(T_{0 \to N}) - \mathbb{E}(T_{0 \to N}^{prior}) = (N-1) * (\frac{1}{2p-1} - \frac{2}{p} + 1) > 0$.*

The proof is deferred to Appendix C. The exploration policy substantially benefits from the involvement of key states, e.g., $\mathbb{E}(T_{0 \to N}) - \mathbb{E}(T_{0 \to N}^{prior}) \to \infty$ with $p \to 0.5$. Such a concept is also commonly seen in practical scenarios, such as in-game checkpoints (Demaine et al., 2016) and landmarks in navigation (Becker et al., 1995).

## 4.2 Key States Localization with LLM

To reduce manual workload, we employ LLM to localize key states. Although generating the aforementioned symbolic key states can be straightforward, LLM's weakness in comprehending symbolic states or environment details necessitates additional information in certain tasks and can lead to errors and hallucinations that are difficult to detect. Here, we stress *the importance of LLM's discriminative ability to localize key states in rollout trajectories* to better leverage LLM's general knowledge. The rationale is that discrimination demands only a high-level task understanding and is more reliable and universal than naive generation, as discussed in detail in Appendix B.1.

To discriminate key states, we prompt LLM to generate $m$ (determined by LLM) discriminator functions $\{\mathcal{F}_i\}_{i=1}^m$, as depicted in Fig. 2. Each discriminator function $\mathcal{F}_i$ (the 'IsKeystate$i$(s)' block in Fig. 2) takes in the state $s_t$ at timestep $t$ and outputs a boolean value to tell whether $s_t$ is the corresponding key state $\kappa_i$. Such an approach systematically annotates each state in trajectories as a key state instance ($s_t \to s_t^{\kappa_2}$ in Fig. 2) or not. Notably, LEMAE injects task-relevant information into the symbolic states without predefined components such as observation captioners (Du et al., 2023) or environment codes (Xie et al., 2023), which require manual fine-tuning, may be unavailable in many scenarios, or could introduce extra information. In addition, the discriminator functions' reusability avoids frequent calls, and our method empirically requires fewer than three LLM inferences for a

specific task. These advantages highlight the potential of LEMAE to expand the scope of application scenarios with fewer constraints and reduced costs.

We design prompts to alleviate the burden of labor-intensive prompt engineering across tasks. As illustrated in Fig. 2, each task's prompt is structured by a standardized prompt template and task information. The prompt template, consistent across tasks, primarily contains several role instructions to guide LLM in role understandings (including **promoting labor division among agents in MARL**) and output constraints, e.g., in JSON format. For a new task with symbolic state space, the prompt template requires only essential details, i.e., the task description and the state form, which can be easily extracted from the task document without additional processing, making it less demanding than previous methods (Ma et al., 2023; Du et al., 2023). LLM then generates key states definitions and discriminator functions, which can be easily extracted from the JSON response via key-value matching. An extension to vision-based tasks is described in Appendix F.2.

Considering that LLM sometimes generates inaccurate responses and non-executable codes, we develop a Self-Check mechanism to enable LLM's autonomous evaluation and response improvement, which is inspired by recent approaches (Shinn et al., 2023; Dhuliawala et al., 2023). The mechanism comprises two checking operations: LLM rethinking and code verification. The former prompts LLM with a set of queries for self-assessment, ensuring compliance with specified criteria. The latter verifies the executability of discriminator functions with actual state inputs, providing feedback until all functions are executable. Table 1 further examines its effectiveness.

We use GPT-4-turbo from OpenAI API and prompt and response details are attached in Appendix D.

### 4.3 KEY STATE-GUIDED EXPLORATION

#### 4.3.1 SUBSPACE-BASED HINDSIGHT INTRINSIC REWARD

With the annotated key states, trajectories can naturally be segmented into sub-trajectories. Drawing inspiration from Andrychowicz et al. (2017), we integrate hindsight intrinsic rewards by conceptualizing the annotated key states as sub-trajectories' subgoals, which is further discussed in Appendix E.3. Such integration guides the policy toward achieving these key states by increasing reward density, thus reducing manual reward design burdens. Moreover, the state vector index from the discriminator function constitutes the reward-related subspace of the state (Liu et al., 2021). Here, we write the Subspace-based Hindsight Intrinsic Reward (SHIR) function as:

$$r_I^{\kappa_m}(t) = \|\Phi_m(s_t) - \Phi_m(\kappa_m)\| - \|\Phi_m(s_{t+1}) - \Phi_m(\kappa_m)\|, \tag{1}$$

where $\|\cdot\|$ denotes a distance metric, e.g., Manhattan distance or cosine similarity; $\Phi_m(s) = (s_e)_{e \in \upsilon_m}$ restricts the state space to elements $e \in \upsilon_m$, $s_e$ is the $e$-th element of the full-state $s$, and $\upsilon_m \subset \mathbb{N}^+$ refers to the subset of entire state space from the discriminator function $\mathcal{F}_m$.

Given that rewards generally rely on a limited subset of the entire state space (Liu et al., 2021; Todorov et al., 2012), adopting subspace-based rewards helps avoid the potential redundancy and bias associated with the design of intrinsic rewards in the entire state space. LEMAE is also applicable to scenarios where rewards depend on the global state space, as it imposes no strict constraints. Hence, the final reward function is further derived as:

$$r(t) = \alpha \cdot r_E(t) + \beta \cdot r_I^{\kappa_m}(t), \tag{2}$$

where $r_E$ denotes the extrinsic reward with $\alpha, \beta \in \mathbb{R}^+$ non-negative scaling factors.

#### 4.3.2 KEY STATES MEMORY TREE

To organize exploration with memory, we introduce the concept of Key States Memory Tree (KSMT). It tracks transitions between key states and further serves exploration and planning. Compared with the naive $\epsilon$-greedy method, gradually revealing the KSMT helps avoid redundant exploration throughout the state space, revealing its potential for real-world applications. Notably, LEMAE is compatible with other memory structures, such as Directed Acyclic Graphs.

**Construct KSMT:** Initialized at the root node, KSMT dynamically expands by iteratively incorporating key state chains obtained from annotated trajectories, as outlined in Algorithm 2. These steps repeat until either reaching the success state or fully depicting the transitions between key states.

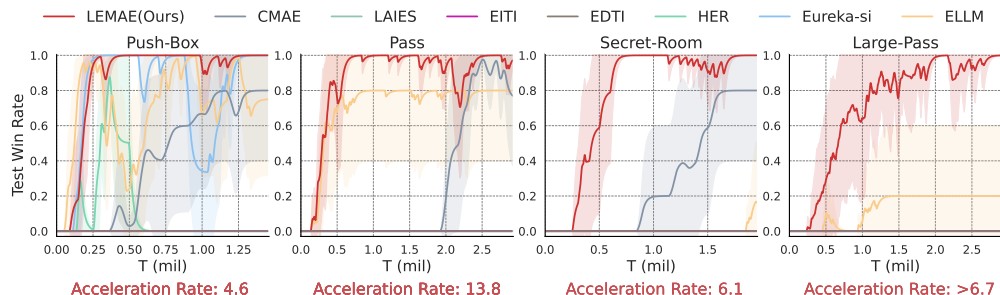

Figure 3: Evaluating LEMAE against baseline methods on four MPE maps with **sparse rewards**, using test win rate as the evaluation metric. The acceleration rate refers to how much faster LEMAE finds the success state compared to CMAE.

**Explore with KSMT:** To discover new KSMT branches, we adopt an exploration strategy that balances high-randomness policy $\pi_\theta^{\epsilon_h}$ for exploring under-explored nodes with low-randomness policy $\pi_\theta^{\epsilon_l}$ to minimize interference with policy learning, as shown in Algorithm 2. Upon reaching a leaf node, agents execute $\pi_\theta^{\epsilon_h}$ to deepen KSMT. While reaching a non-leaf node $\xi_i$, the agents take $\pi_\theta^{\epsilon_h}$ with probability $p_i$ to expand the breadth or $\pi_\theta^{\epsilon_l}$ with probability $1 - p_i$ for progression towards the next key state. The probability $p_i$ is calculated as $p_i = \frac{1}{d_i+1}$, with $d_i$ the degree of the node $\xi_i$ as an indicator of the degree of under-exploration. The exploration phase completes upon the discovery of the success state. We also prune branches that do not lead to success to circumvent task-irrelevant key states. In this way, KSMT enables exploration in a more meaningful state subspace.

**Plan with KSMT:** Since KSMT acts as a dynamic model within the key state space, we plan the subgoal for the final sub-trajectory based on it. As shown in Fig. 2b, given the achieved key states chain, we identify the corresponding branch ($\kappa_2 \to \kappa_1 \to$ children) in KSMT through a lookup operation. Since they have been validated by memory, the children represent the most likely next key states, from which we randomly sample the final subgoal. This process mainly handles cases where trajectories fail to reach a key state as the final subgoal. It enhances SHIR and improves the efficacy of exploring KSMT by encouraging agents to access existing key states.

## 5 EXPERIMENTS

We conduct experiments on commonly used multi-agent exploration benchmarks: (1) the Multiple-Particle Environment (Lowe et al., 2017; Wang et al., 2019) and (2) the StarCraft Multi-Agent Challenge (Samvelyan et al., 2019b). Following previous studies (Ma et al., 2023; Liu et al., 2021; Xu et al., 2023), we focus primarily on tasks with symbolic state spaces and use the **sparse reward** version for all tasks without specific instructions.

**Baselines.** We compare LEMAE with representative baselines: **IPPO** is a MARL algorithm which extends PPO (Schulman et al., 2017); **QMIX** (Rashid et al., 2018) is a widely adopted MARL baseline; **EITI** and **EDTI** (Wang et al., 2019) employ the impact of interaction in coordinated agents' behaviors; **MAVEN** (Mahajan et al., 2019) combine value-based and policy-based approaches through hierarchical control; **CMAE** (Liu et al., 2021) learns cooperative exploration by selecting shared goals from multiple projected state space; **RODE** (Wang et al., 2020b) decomposes joint action spaces into role-based ones to enhance exploration; **MASER** (Jeon et al., 2022) generates subgoals automatically for multiple agents from the experience replay buffer; **LAIES** (Liu et al., 2023) addresses the lazy agents problem by mathematical definition and causal analysis. **ELLM** (Du et al., 2023) employs LLM priors to guide vision-based exploration, using state captioners and semantic similarity-based rewards. LEMAE is implemented on **IPPO** in MPE and **QMIX** in SMAC, consistent with previous works (Wang et al., 2019; Liu et al., 2023; Jeon et al., 2022) to ensure fair comparisons.

We run each algorithm on five random seeds and report the mean performance with standard deviation. Further details can be referenced in Appendix E.

### 5.1 MULTIPLE-PARTICLE ENVIRONMENT (MPE)

In MPE, we evaluate LEMAE on *Pass*, *Secret-Room*, *Push-Box*, and *Large-Pass*, which are commonly used multi-agent exploration tasks in previous works (Wang et al., 2019; Liu et al., 2021).

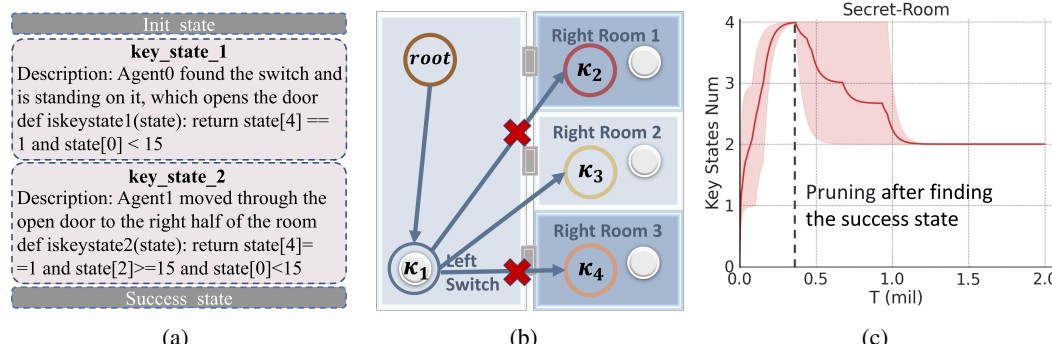

Figure 4: (a) Key states discrimination functions generated on task *Pass*. (b) The map of *Secret-Room* with key states: $\kappa_1$ represents occupying the left switch to open all doors, while $\kappa_2$, $\kappa_3$, and $\kappa_4$ represent exploring right rooms 1, 2, and 3, respectively. The directional arrows symbolize the transitional relationships within KSMT. (c) The key states number curve in *Secret-Room* shows that LEMAE can identify all key states and proficiently prune task-irrelevant ones.

**LLM can effectively discriminate key states.** To start with, we examine the efficacy of LLM in discriminating key states. On the *Pass* task, as shown in Fig. 1a, a room is divided by a wall, each half containing an invisible switch. Passage through the door is allowed only when an agent occupies a switch. Initially, in the left half-room, agents must cooperate to move to the right half-room. In Fig. 4a, LLM exhibits a precise understanding of the task and generates meaningful discriminator functions, demonstrating the feasibility of our approach based on the current LLM.

**LEMAE achieves superior performance.** We investigate how LEMAE enhances exploration by comparing it with baselines, confirming the value of incorporating LLM priors. The training curves are depicted in Fig. 3. The failure of commonly used baselines highlights the necessity and urgency for efficient exploration, while the superior performance of LEMAE underscores the effectiveness of augmenting RL with task-specific guidance from LLM. Specifically, the failure of EITI, EDTI, and LAIES may be attributed to the complexity of learning dynamics or the scarcity of external state changes in the tasks. While CMAE learns effective strategies for simple tasks, its redundant exploration hampers efficiency, rendering it inadequate for tasks with expansive exploration spaces, such as *Large-Pass*. Although it benefits from LLM priors, ELLM performs worse than LEMAE due to the weak guidance provided by semantic similarity-based rewards, not to mention its reliance on frequent LLM inference and a predefined state captioner. Furthermore, we compare LEMAE with traditional SOTA baseline CMAE using the metric of the number of exploration steps taken to find the success state. The results indicate a significant exploration **acceleration rate, up to 10x**, underscoring LEMAE's efficiency. The superior performance of our method can be attributed to the mitigating of redundant exploration by incorporating task-relevant information.

**LEMAE benefits from LLM priors through discrimination.** We evaluate **HER** (Andrychowicz et al., 2017), which also employs hindsight intrinsic rewards but selects goals randomly from memory. HER's poor performance emphasizes the critical role of incorporating LLM priors for localizing key states in achieving efficient exploration. To support our claim about the superiority of LLM discrimination over generation, we evaluate **Eureka-si**, a single-iteration Eureka (Ma et al., 2023) variant, which uses LLM to generate reward functions. While Eureka-si performs comparably to LEMAE in simple tasks, it struggles in complex tasks with partial observability, indicating that LLM-based discrimination may offer a more general and effective integration of LLM. Notably, these two methods are not specifically designed for efficient exploration. To ensure fairness, we retain the prompt information consistent across all relevant LLM-based methods. Please refer to Appendix E.2, E.3 for details.

**LEMAE reduces redundant exploration.** We further compare the exploration behavior of LEMAE with that of CMAE on the *Pass* task. The visitation maps, displayed in log scale, are depicted in Fig. 1. The illustration reveals that LEMAE markedly avoids redundant exploration: agents trained with CMAE tend to excessively explore the left room, while the agents' visitation area in LEMAE is notably concentrated around the success path. Furthermore, an organic division of labor among agents emerges, affirming the efficacy of encouraging labor division in prompt design.

**LEMAE circumvents task-irrelevant key states.** Due to the incomplete information, LLM may discriminate task-irrelevant key states. For instance, in the *Secret-Room* task, three rooms are present

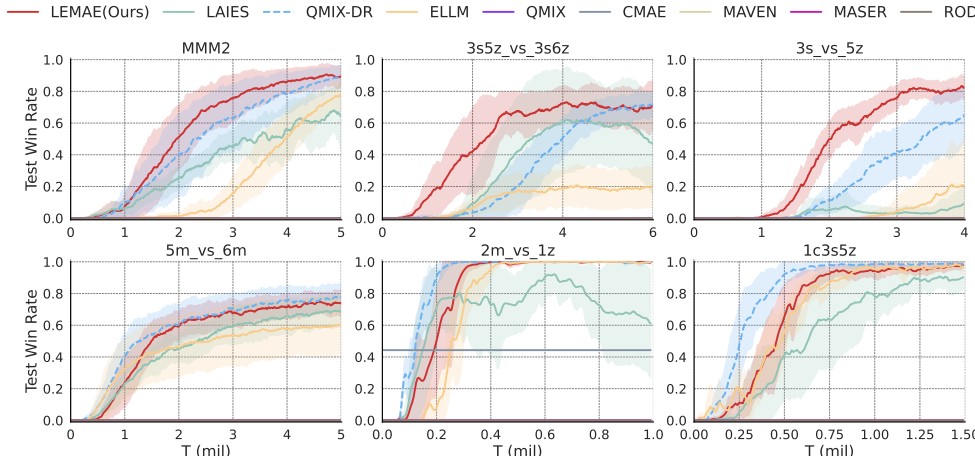

Figure 5: Evaluating LEMAE on six SMAC maps with **sparse rewards**, using test win rate as the evaluation metric. Notably, **QMIX-DR** is QMIX with dense rewards in the original SMAC, serving as an upper bound, with its intrinsic rewards designed by a human expert.

on the right, but LLM is not informed about the real target room for fairness. In Fig. 4b, LLM discriminates two task-irrelevant key states, denoted as $\kappa_2$ and $\kappa_4$, which represent an exploration of the two irrelevant rooms, respectively. Fig. 4c shows that the pruning mechanism after finding the success state in LEMAE makes it effective in circumventing task-irrelevant key states. A more detailed robustness analysis is provided in Sec. 5.5.

## 5.2 STARCRAFT MULTI-AGENT CHALLENGE (SMAC)

SMAC is a widely-used challenging benchmark in MARL. In contrast to dense or semi-sparse reward versions used before, we employ **fully sparse-reward tasks** to emphasize exploration, rewarding agents only upon complete enemy elimination. In addition, to validate LEMAE across diverse scenarios, we conduct experiments on six maps with varied difficulty and agent numbers.

In Fig. 5, LEMAE demonstrates superior performance over all baselines. Although baselines QMIX, MAVEN, CDS, and MASER excel in dense or semi-sparse reward settings, they struggle in fully sparse reward scenarios. CMAE shows partial efficacy in simpler tasks but fails in harder scenarios due to the lack of task-related information in curiosity-driven goal selection. LAIES is the only non-LLM baseline comparable to LEMAE. However, it requires handcrafted external state priors and still underperforms compared to LEMAE, especially on more challenging tasks. ELLM, benefiting from LLM priors, performs well on simpler tasks, but its effectiveness diminishes on harder ones, likely due to the instability and less reliable guidance of semantic similarity-based rewards. Notably, we add **QMIX-DR**, which augments QMIX with dense rewards in the original SMAC. Surprisingly, LEMAE demonstrates the potential to match or even surpass QMIX-DR, particularly in hard maps, shedding light on minimizing the manual workload in complex reward design in real-world scenarios. Given the complexity of the SMAC benchmark, the consistent superiority of LEMAE confirms its potential applicability in more complex real-world scenarios. We further evaluate LEMAE on SMACv2 (Ellis et al., 2024), an enhanced version with more stochasticity, as detailed in Appendix F.1.

## 5.3 COMPATIBLILITY WITH VARIOUS ALGORITHMS

LEMAE incorporates task-relevant guidance in the form of intrinsic rewards and is agnostic to RL algorithms. Sec. 5.1 and 5.2 have verified the compatibility through implementing on two distinct MARL algorithms: **IPPO** in MPE and **QMIX** in SMAC. To further substantiate this claim, we build our method on two widely-used MARL algorithms, namely **QPLEX** (Wang et al., 2020a) and **VMIX** (Su et al., 2021), adopting a value-based and actor-critic methodology respectively. As illustrated in Fig. 6a, algorithms combined with LEMAE consistently improve performance, underscoring the potential of LEMAE to integrate with alternative algorithms across diverse fields in the future. Additionally, LEMAE is a versatile approach for efficient exploration, not limited to

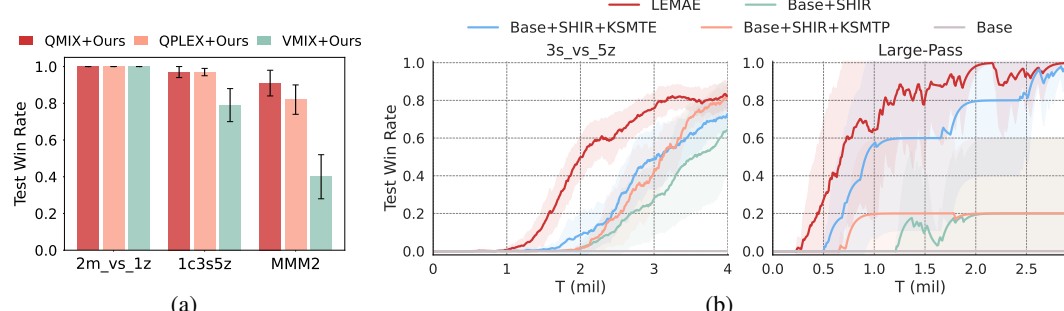

Figure 6: (a) Evaluations on baselines of well-known MARL algorithms, i.e., **QPLEX** and **VMIX**. Notably, both QPLEX and VMIX exhibit complete failure unless integrated with our approach. (b) Ablation studies are conducted on two exemplary tasks from MPE and SMAC to assess the significance of KSMT and SHIR within LEMAE.

Table 1: Ablation studies on Self-Check mechanism and LLMs. We compare the performance of two LLMs (GPT-4-turbo and GPT-3.5-turbo), recording the Acceptance Rate ($r_{acc}$) and Execution Rate ($r_{exe}$) in ten runs of the generated discriminator functions. w/o denotes the absence of our Self-Check mechanism.

| $r_{acc}$ ($r_{exe}$) | GPT-4-turbo | GPT-4-turbo w/o | GPT-3.5-turbo |
|---|---|---|---|
| Large-Pass | **1.0** (1.0) | 0.8 (1.0) | 0.7 (1.0) |
| 2m_vs_1z | **1.0** (1.0) | 0.7 (1.0) | 0.6 (1.0) |
| 5m_vs_6m | **1.0** (1.0) | 0.9 (1.0) | **1.0** (1.0) |
| MMM2 | **0.8** (1.0) | 0.6 (0.7) | 0.0 (1.0) |

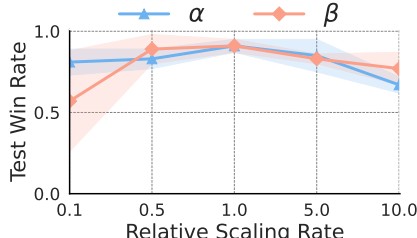

Figure 7: Hyperparameter examination on reward scaling rate $\alpha$ and $\beta$. The x-axis represents the relative values with respect to the default parameters.

MARL. To validate this assertion, we conduct further evaluations of LEMAE in a single-agent variant of MPE, as demonstrated in Appendix F.4.

## 5.4 ABLATION STUDIES

**Role of SHIR and KSMT.** We conduct an ablation study to assess the significance of KSMT and SHIR within LEMAE. We select two exemplary tasks from MPE and SMAC and report results in Fig. 6b. In SMAC, Base refers to QMIX, while in MPE, it denotes IPPO. Besides, SHIR represents subspace-based hindsight intrinsic reward, KSMTE signifies exploration with KSMT, KSMTP denotes planning with KSMT, and LEMAE encompasses Base+SHIR+KSMTE+KSMTP. As illustrated, the absence of SHIR or KSMT significantly deteriorates performance, revealing both components' pivotal roles in achieving effective key state-guided exploration.

**Role of Self-Check mechanism and LLMs.** We conduct a comparative analysis between GPT-4-turbo and GPT-3.5-turbo regarding generating discriminator functions. Meanwhile, we investigate the performance of GPT-4-turbo without the Self-Check mechanism (GPT-4-turbo w/o). The Acceptance Rate ($r_{acc}$) denotes the proportion of seeds achieving over $80\%$ of the best performance after RL training, while the Execution Rate ($r_{exe}$) indicates the proportion of seeds for which all discriminator functions are executable. As depicted in Table 1, the results demonstrate that a powerful LLM with our Self-Check mechanism effectively ensures the high quality of key states, as evidenced by the code's executability and the final performance. The scalability of LEMAE to LLM and our Self-Check mechanism promise that LEMAE can leverage more powerful LLMs in the future and be applied to more challenging real-world tasks safely and efficiently.

## 5.5 SENSITIVITY & ROBUSTNESS ANALYSIS

**Sensitivity to Hyperparameters.** We conduct experiments on the pivotal hyperparameters in LEMAE, i.e., reward scaling rates $\alpha$ and $\beta$. The x-axis represents the relative values with respect

to the default ($\alpha = 10$, $\beta = 1$), encompassing evaluations for $\alpha \in \{1, 5, 10, 50, 100\}$ and $\beta \in \{0.1, 0.5, 1, 5, 10\}$. Fig. 7 illustrates that LEMAE is robust to these hyperparameters across a considerable range. Notably, excessive extrinsic reward scaling rate $\alpha$ or insufficient intrinsic reward scaling rate $\beta$ can cause performance degradation due to the abrupt alteration of the reward or the inadequate motivational impact. Additionally, we conduct an ablation study on mixed-randomness exploration in Appendix F.3.

**Robustness to Perturbations in Key States.** We conduct experiments to evaluate the robustness of LEMAE to perturbations in key states. Specifically, Reduction simulates the absence of key states by randomly clearing a certain percentage of key states. Distraction simulates the misidentification of common states as key states by randomly adding a certain portion of distracting states (encouraging a random state dimension to 0). The observed performance decrease with increasing perturbations in Table 2 underscores the significance of key states' quality. LEMAE exhibits notable robustness to perturbations, ensuring its reliability across diverse application scenarios, particularly in light of the limited capabilities of current LLMs.

Table 2: Robustness analysis of LEMAE to perturbations in key states, whether randomly deleting key states (Reduction) or adding distracting states (Distraction).

| Tasks | Default | Reduction | | Distraction | |
|---|---|---|---|---|---|
| | | 25% | 50% | 50% | 100% |
| 1c3s5z | **0.98**±**0.02** | 0.97±0.01 | 0.97±0.02 | 0.92±0.04 | 0.89±0.05 |
| 3s_vs_5z | **0.83**±**0.07** | 0.80±0.18 | 0.57±0.28 | 0.80±0.11 | 0.66±0.08 |
| MMM2 | **0.89**±**0.08** | **0.89**±**0.03** | 0.79±0.09 | 0.86±0.04 | 0.79±0.08 |

## 5.6 SCALABILITY & GENERALIZATION ANALYSIS

To rule out the possibility that LEMAE's success relies on LLM's familiarity with the chosen tasks, we've handcrafted a brand new task, termed *River*, which LLM has never encountered before. The task is illustrated in Fig. 8a, where the objective is for Bob to help Alice, who is afraid of water, cross two rivers to reach the bottom-right corner. As shown in Fig. 8b, LEMAE outperforms the baselines, and this confirms LLM's generalization capabilities to empower LEMAE's effectiveness in promoting efficient exploration in diverse new tasks. Please refer to Appendix E.4.3 for details on the task.

Additionally, we extend LEMAE to a vision-based task, as described in Appendix F.2, demonstrating the scalability potential of LEMAE.

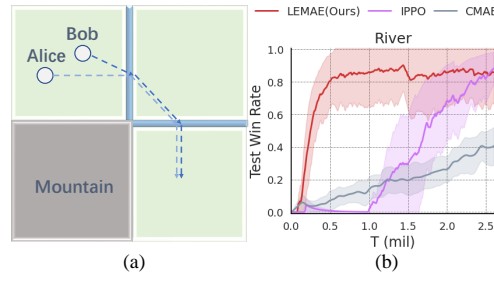

Figure 8: (a) A brand new task, *River*, which LLM has never encountered before. (b) The training curves of LEMAE and baselines using the evaluation metric of test win rate.

## 6 CONCLUSION

**Summary of This Work:** We present LEMAE, a novel framework that benefits multi-agent exploration with task-specific guidance from LLM. LEMAE executes the *key states localization with LLM* and enables the *key state-guided exploration* to improve sample efficiency. In this way, we can (i) build up connections between LLM and RL to ground linguistic knowledge into decision-making, (ii) reduce the manual workload in accessing knowledge and intensive inference calls from LLM, and (iii) significantly boost exploration efficiency through guided and organized exploration. Extensive experiments further examine the effectiveness of LEMAE in typical benchmarks.

**Limitations & Future Investigations:** In developing LEMAE, we made efforts to compensate for the pitfalls of concurrent LLMs, e.g., careful preparation for prompt engineering and task-related prior provision to avoid the nuisances in LLM usages. All of these can be circumvented with the progress of LLM's capability enhancement. This work paves the way for LLM-empowered RL to achieve the potential in complicated decision-making scenarios.

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

# A  ALGORITHM

This section includes the pseudo algorithms. Algorithm 1 presents LEMAE's main algorithm. LEMAE consists of four phases: generating discriminator functions with LLM, exploring with KSMT, calculating SHIR, and performing RL training. For on-policy RL, the buffer $D$ corresponds to a rollout buffer, while for off-policy RL, it is initialized as a replay buffer (Paischer et al., 2022). Algorithm 2 illustrates the process of exploring with KSMT. As our approach is agnostic to reinforcement learning algorithms, we leave out the details of standard RL training in the main paper.

---

**Algorithm 1:** LEMAE

---

**Input:** Large language model $\mathcal{M}$, prompt $\mathcal{P}$, rethinking prompt $\mathcal{P}^{re}$, non-negative scaling
  factors $\alpha, \beta$, randomness epsilon $\epsilon_l, \epsilon_h$ ($\epsilon_l < \epsilon_h$), training frequency $\mathcal{N}$, max episodes
  $\mathcal{N}^{max}$, key states numbers $\mathcal{K}$.

**Output:** Policy network $\pi_\theta$.

Randomly initialize the policy network parameter $\theta$.

Initialize key states memory tree $\mathcal{T} \leftarrow [root]$, replay buffer $\mathcal{D}$, key states chain replay buffer $\mathcal{D}^{ks}$

Initial discriminator functions $\{\hat{\mathcal{F}}_i\}_{i=1}^{\mathcal{K}} \leftarrow \mathcal{M}(\mathcal{P})$;

// Self-Check Mechanism

LLM rethinking $\{\hat{\mathcal{F}}_i^{re}\}_{i=1}^{\mathcal{K}} \leftarrow \mathcal{M}(\mathcal{P}, \{\hat{\mathcal{F}}_i\}_{i=1}^{\mathcal{K}}, \mathcal{P}^{re})$;

**while** *there are non-executable discriminator functions in* $\{\hat{\mathcal{F}}_i^{re}\}_{i=1}^{\mathcal{K}}$ **do**

 $\quad \{\hat{\mathcal{F}}_i^{re}\}_{i=1}^{\mathcal{K}} \leftarrow \mathcal{M}(\mathcal{P}, \{\hat{\mathcal{F}}_i^{re}\}_{i=1}^{\mathcal{K}}, error)$

**end**

Final discriminator functions $\{\mathcal{F}_i\}_{i=1}^{\mathcal{K}}$;

**for** $episode = 1$ **to** $\mathcal{N}^{max}$ **do**

 $\quad$ // Explore with Key States Memory Tree (Algorithm 2)

 $\quad \kappa\_chain, \mathcal{T}, \tau \leftarrow$ KSMT-Exp $(\pi_\theta, \mathcal{T}, \{\mathcal{F}_i\}_{i=1}^{\mathcal{K}}, \epsilon_l, \epsilon_h)$;

 $\quad \mathcal{D} \leftarrow \mathcal{D} \cup \{\tau\}, \mathcal{D}^{ks} \leftarrow \mathcal{D}^{ks} \cup \{\kappa\_chain\}$;

 $\quad$ **if** $episode \bmod \mathcal{N} = 0$ **then**

 $\quad\quad$ Sample a batch $B = \{\tau_i\}_{i=1}^{|B|}$ from $\mathcal{D}$ and the corresponding batch of key states chains

 $\quad\quad B^{ks} = \{\kappa\_chain_i\}_{i=1}^{|B|}$ from $\mathcal{D}^{ks}$;

 $\quad\quad$ **for** $\tau = \{(s_t, \boldsymbol{a}_t, s_{t+1}, r_t)\}_{t=1}^{t_{max}}, \kappa\_chain \in B, B^{ks}$ **do**

 $\quad\quad\quad t^{start} \leftarrow 1$;

 $\quad\quad\quad$ **for** $\{t^{end}, \kappa_m\} \in \kappa\_chain$ **do**

 $\quad\quad\quad\quad$ **for** $t = t^{start}$ to $t^{end}$ **do**

 $\quad\quad\quad\quad\quad$ // Subspace-based Hindsight Intrinsic Reward (equation 1)

 $\quad\quad\quad\quad\quad$ Update $\tau$ with $r_t = \alpha \cdot r_t + \beta \cdot r_I^{\kappa_m}(s_t, s_{t+1}); t^{start} \leftarrow t^{end}$;

 $\quad\quad\quad\quad$ **end**

 $\quad\quad\quad$ **end**

 $\quad\quad\quad$ // Plan with Key States Memory Tree

 $\quad\quad\quad$ **if** *branch corresponding to* $\kappa\_chain$ *in* $\mathcal{T}$ **then**

 $\quad\quad\quad\quad$ Extract all children nodes $\mathcal{C} = \{\xi_i\}_{i=1}^{|\mathcal{C}|}$ and randomly sample a node $\xi_i \sim \mathcal{C}$;

 $\quad\quad\quad$ **end**

 $\quad\quad\quad$ **else**

 $\quad\quad\quad\quad$ Randomly sample a node $\xi_i$ from all nodes (not in $\kappa\_chain$) in $\mathcal{T}$;

 $\quad\quad\quad$ **end**

 $\quad\quad\quad \kappa_{plan}$ corresponds to the sampled node $\xi_i$;

 $\quad\quad\quad$ **for** $t = t^{start}$ to $t^{max}$ **do**

 $\quad\quad\quad\quad$ Update $\tau$ with $r_t = \alpha \cdot r_t + \beta \cdot r_I^{\kappa_{plan}}(s_t, s_{t+1})$;

 $\quad\quad\quad$ **end**

 $\quad\quad\quad$ Update $B$ with $\tau$;

 $\quad\quad\quad$ // RL Training (Algorithm Agnostic)

 $\quad\quad\quad$ Use an MARL algorithm to train $\theta$ with $B$;

 $\quad\quad$ **end**

 $\quad$ **end**

**end**

**Algorithm 2:** Explore with Key States Memory Tree (KSMT-Exp)

---

**Input:** Policy network $\boldsymbol{\pi}_\theta$, key states memory tree $\mathcal{T}$, discriminator functions $\{\mathcal{F}_i\}_{i=1}^{\mathcal{K}}$,
        randomness epsilon $\epsilon_l, \epsilon_h$ ($\epsilon_l < \epsilon_h$).
**Output:** $\kappa\_chain$, key states memory tree $\mathcal{T}$, trajectory $\tau$.
Initialize $\kappa\_chain \leftarrow [\ ], \tau \leftarrow \{\}$
**for** $t = 1$ **to** $t_{\max}$ **do**
    // key states localization in rollout trajectories
    Discriminate $s_t$ with $\{\mathcal{F}_i\}_{i=1}^{\mathcal{K}}$;
    **if** *($s_t$ is a key state $\kappa_m$) and ($\kappa_m$ not in $\kappa\_chain$)* **then**
        $\kappa\_chain$.append($\{t, \kappa_m\}$);
        **if** *branch corresponding to $\kappa\_chain$ not in $\mathcal{T}$* **then**
            // update KSMT
            add the branch into $\mathcal{T}$;
        **end**
        **else if** *$\kappa_m$ correponds to a non-leaf node $\xi$* **then**
            // mixed-randomness exploration strategy
            $d \leftarrow$ the degree of the node $\xi$;
            $p = \frac{1}{d+1}$;
            $\epsilon = \begin{cases} \epsilon_h & \text{with probability } p \\ \epsilon_l & \text{with probability } 1-p \end{cases}$;
        **end**
        **else**
            $\epsilon = \epsilon_h$;
        **end**
    **end**
    With probability $\epsilon$ select a random action $\boldsymbol{a}_t$;
    Otherwise select $\boldsymbol{a}_t \sim \boldsymbol{\pi}_\theta(s_t)$;
    Obtain a tuple $(s_t, \boldsymbol{a}_t, s_{t+1}, r_t)$ by executing $\boldsymbol{a}_t$;
    $\tau \leftarrow \tau \cup \{(s_t, \boldsymbol{a}_t, s_{t+1}, r_t)\}$
**end**

---

## B  FURTHER DISCUSSIONS

### B.1  THE INSIGHTS BEHIND KEY STATES DISCRIMINATION

In our considered scenarios, we claim that discrimination is generally easier and more universal than key state generation by LLM, particularly in the context of high-dimensional states and partial observability. The reasons are as follows:

1. Discrimination focuses on high-level task understanding and identifying key state characteristics, while generation requires detailed, low-level comprehension, assigning values to each element. This makes generation more challenging and error-prone, particularly in high-dimensional settings. Discrimination equivalently simplifies the output space to key state labels, thus alleviating issues like hallucinations.

2. In implementations, errors in discriminator functions are easier to examine and correct through testing with real states. In contrast, errors in generated key states are harder to detect and are typically inferred from training performance.

3. In cases of partial observability, generating key states directly is unreliable. For example, in the *Pass* task, the positions of hidden switches are unknown and must be inferred from the door's status. LLM cannot generate key states accurately without knowledge of the specific agents' positions required to activate a switch.

### B.2  LIMITATIONS

We build a bridge between LLM and RL to facilitate efficient exploration by leveraging task-related guidance provided by LLM. However, persistent constraints inherent to LLMs, such as their limited

capacity to comprehend task-specific information and the inevitable hallucination, become bottlenecks in our approach, which induces the following limitations:

1. We mitigate heavy prompt engineering through the use of a standardized prompt template but the necessity persists for manually providing task information for LLM. Thus, we assume the availability of semantic meanings for symbolic states. This assumption is feasible, as these manually designed states have inherent meanings documented in task specifications (Liu et al., 2021; Samvelyan et al., 2019b), and is no stronger than prior works requiring a state captioner (Du et al., 2023) or environment code (Ma et al., 2023), which also involve manual fine-tuning or access to additional state information.

2. Constrained by the limitations of LLM's capabilities and the inherent issue of hallucination, it may face challenges in directly providing effective key state priors for more complex tasks. Besides, due to our use of LLMs, this work primarily focuses on tasks with symbolic states. Future research could extend its application to more complicated tasks, e.g., image-based tasks, by employing advanced multi-modal LLMs. We provide an initial attempt to extend LEMAE beyond symbolic tasks in Appendix F.2.

Since the efficiency of the proposed LEMAE is essentially derived from versatile LLMs, we believe that the surge of foundation model exploration will flourish LLM-empowered RL.

## B.3 FUTURE WORKS

The success of the proposed LEMAE highlights the necessity and efficacy of empowering RL with LLM. To enhance performance and extend applicability, we will explore two avenues for future research aimed at addressing the identified limitations. These avenues are outlined as follows:

1. Streamlining the task information provision through multi-modal self-collection: Multi-modal LLMs are garnering increasing attention for their ability to comprehend situations through various modalities. Incorporating them with self-exploration and memory mechanisms shows promise in automating the collection and understanding of task information, thereby streamlining the implementation and enhancing the adaptability of LEMAE. We provide an initial attempt to extend LEMAE beyond symbolic tasks in Appendix F.2.

2. Unleashing the power of better LLM with an iterative feedback mechanism: Undoubtedly, given the rapid pace of LLM development, the emergence of more powerful LLMs is imminent. On one hand, we intend to harness the capabilities of these advanced LLMs. On the other hand, to fully unleash the potential of LLMs, we plan to devise an iterative feedback mechanism to feedback LLM in LEMAE during RL training to mitigate issues like hallucinations and errors in task understanding.

## B.4 BROADER IMPACTS

Large Language Models have demonstrated considerable potential in showcasing impressive capabilities across various downstream tasks. However, research on empowering RL with LLMs is still nascent. As a pioneering endeavor to empower RL with LLM, we propose a general approach facilitating efficient exploration in RL with task-specific guidance from LLM.

1. For the research community, the publication of this work will inspire further exploration into encouraging the integration of LLMs with RL to address the inherent challenges in RL, such as efficient exploration, limited sample efficiency, and unsatisfactory generalization. Additionally, our design promotes the application of discrimination and coding to ground linguistic knowledge from LLMs into symbolic tasks.

2. LEMAE shows promise for real-world deployment in scenarios requiring efficient exploration, such as autonomous vehicle control and robot manipulation. Moreover, as LLM is growing by leaps and bounds, it is foreseeable that LEMAE can be applied to more challenging real-world tasks by taking advantage of more powerful LLM. Notably, to mitigate potential risks, it is imperative to conduct LLM generation and RL training under human supervision, thereby ensuring undesirable outcomes are averted.

# C Proof of Proposition 4.1

*Proof.* Random walk is a fundamental stochastic process, formed by successive summation of independent, identically distributed random variables (Lawler & Limic, 2010). This work considers the one-dimensional asymmetric random walk problem, where an agent starts at $x = 0$ and aims to reach $x = N \in \mathbb{N}^+, N > 1$. The expected first hitting time considered as the metric of performance, implying the average computational time complexity (Yu & Zhou, 2008). Below is the proof of Proposition 4.1.

Firstly, we can prove the expected first hitting time within the default setting through the application of martingale theory. According to the problem setting, we can define the movement at each time step as: $M_0 = 0, M_1, M_2...$ are i.i.d. random variables with distribution $P(M_i = 1) = p, P(M_i = -1) = 1 - p, p \in (0.5, 1)$. Then the position of agent after n steps can be represented as:

$$S_n = \sum_{i=1}^{n} M_i, S_0 = 0 \tag{3}$$

However, because of the asymmetry of random variables $M_i$, $\{S_n, n \geq 0\}$ does not pertain to the martingale $w.r.t.$ $\{M_n, n \geq 1\}$. It's observed that $\mathbb{E}(M_i) = 2p - 1, i \geq 1$. Then, we can define:

$$Y_n = \sum_{i=1}^{n} \left( M_i - (2p - 1) \right), Y_0 = 0 \tag{4}$$

It's easy to prove that

$$\mathbb{E}|Y_n| = \sum_{i=1}^{n} E|M_i| - n(2p - 1) = 2n - 2np < \infty \tag{5}$$

$$\mathbb{E}(Y_{n+1}|M_0, M_1, ...M_n) = Y_n + \mathbb{E}(M_{n+1}) - (2p - 1) = Y_n \tag{6}$$

So, according to the definition, $\{Y_n, n \geq 0\}$ is a martingale $w.r.t.$ $\{M_n, n \geq 1\}$

Let $T_{0 \to N} = min\{n : S_0 = 0, S_n = N\} = min\{n : Y_0 = 0, Y_n = N - n * (2p - 1)\}$. It's clear that $T_{0 \to N}$ is a stopping time $w.r.t.$ $\{M_n, n \geq 1\}$.

It's easy to prove that

$$\mathbb{E}\left(|Y_{n+1} - Y_n||M_0, M_1, ...M_n\right) = \mathbb{E}\left(|M_{n+1}|\right) - (2p - 1) = 2 - 2p < 2 \tag{7}$$

We can assume that $\mathbb{E}(T_{0 \to N}) < \infty$. Then, according to the Optional Stopping Theorem (Durrett, 2019), we can get

$$\mathbb{E}(Y_{T_{0 \to N}}) = N - \mathbb{E}(T_{0 \to N}) * (2p - 1) = \mathbb{E}(Y_0) = 0 \tag{8}$$

Then

$$\mathbb{E}(T_{0 \to N}) = \frac{N}{2p - 1} \tag{9}$$

The assumption $\mathbb{E}(T_{0 \to N}) < \infty$ is thereby validated. Consequently, the expected first hitting time within the default setting is $\mathbb{E}(T_{0 \to N}) = \frac{N}{2p-1}$, a conclusion also articulated in Theorem 4.8.9 of Durrett (2019).

We can introduce the task-relevant information that the agent must first reach key states: $\kappa = 1, ..., N - 1$ before progressing to $x = N$. It is presupposed that every time the agent achieves at $x = \kappa$, the policy where $x < \kappa$ is updated to a deterministic rightward movement, i.e., $P(M_x = 1) = 1, x < \kappa$, thereby emulating the update process in Reinforcement Learning.

The expected first hitting time from $x = 0$ to $x = 1$ is $\mathbb{E}(T_{0 \to 1}) = \frac{1}{2p-1}$. After reaching $x = 1$, the expected first hitting time from $x = 1$ to $x = 2$ can be calculated as:

$$\mathbb{E}(T_{1 \to 2}^{prior}) = p * \sum_{n=0}^{\infty} (2n + 1)(1 - p) = \frac{2}{p} - 1 \tag{10}$$

Similarly, we can easily prove that

$$\mathbb{E}(T_{1 \to N}^{prior}) = (N-1) * \left(\frac{2}{p} - 1\right) \tag{11}$$

Consequently, the expected first hitting time after the integration of priors becomes $\mathbb{E}(T_{0 \to N}^{prior}) = \mathbb{E}(T_{0 \to 1}) + \mathbb{E}(T_{1 \to N}^{prior}) = \frac{1}{2p-1} + (N-1) * \left(\frac{2}{p} - 1\right)$.

The total advantage resulting from the integration of appropriate priors is expressed as $\mathbb{E}(T_{0 \to N}) - \mathbb{E}(T_{0 \to N}^{prior}) = (N-1) * (\frac{1}{2p-1} - \frac{2}{p} + 1) > 0, p \in (0.5, 1), N \in \mathbb{N}^+, N > 1$

$\square$

# D  LLM PROMPTS AND RESPONSES

Here are the example prompt and response in our work. Please reference the code for further details. Notably, we adopt the chain-of-thought technique from Wei et al. (2022).

---

**SMAC Prompt and Response Example**

**SYSTEM:**

**(Task_Description)**
We are playing StarCraft II micro scenario, tring to control our agents to defeat all of the enemy units.

**(State_Form)**
In each step, the current state is represented as a 1-dimensional list:
$[nf\_al] * n\_agents + [nf\_en] * n\_enemies + [last\_actions]$.

$nf\_al$ denotes the unit state for each agent with attributes
$[health\_rate, weapon\_cooldown\_rate, relative\_x\_to\_map\_center,$
$relative\_y\_to\_map\_center, shield\_rate$ (1 dimension if a_race is P else 0 dimension),
$unit\_type\_bits$ (the dimension is defined in the map config)].

$nf\_en$ represents the unit state for each enemy with attributes
$[health\_rate, relative\_x\_to\_map\_center, relative\_y\_to\_map\_center,$
$shield\_rate$ (1 dimension if b_race in map config is P else 0 dimension),
$unit\_type\_bits$ (the dimension is defined in the map config)].

The $last\_actions$ component does not require consideration.

**(Role_Instruction)(Template)**
Your role is to give several critical key states in the task which we should try to reach and generate the corresponding discriminator function for each key state which can discriminate if the input state has reached the key state.
Note:
    1. Don't use the information you are not told.
    2. The code should be as generic as possible.
    3. The discriminator functions for different key states should be independent.
    4. Your answer should be complete and not omitted.
Please think step by step and adhere to the following JSON format (just replace the () with your answer):
{

---

Thought: (Please fully understand the task. Think step by step and give your thought about the concrete state here),
    Key states: {
      init: all agents and enemies are alive,
      key_state_1: (description),
      key_state_2: (description),
      ...,
      success: all enemies are dead and at least one agent is alive
    },
    Discriminators: {
      key_state_1: (a python function with the form of def iskeystate1(state): ... return 0 or 1),
      key_state_2: (a python function with the form of def iskeystate2(state): ... return 0 or 1),
      ...
    }
    Subspace_states: {
      key_state_1: (extract subspace state used in corresponding discriminator functions, such as [1] or [2,5]),
      key_state_2: (extract subspace state used in corresponding discriminator functions, such as [1] or [2,5]),
      ...
    }
}

**USER:**

Task is *2m_vs_1z*. The map config is: {
    n_agents: 2, n_enemies: 1, limit: 150, a_race: T, b_race: P, unit_type_bits: 0, map_type: marines
}.
Give more than 2 key states and the corresponding discriminator functions.

**ASSISTANT: (LLM response)**

{
    Thought: Since a_race is T, there is no shield_rate in nf_al; b_race is P, so there is 1-dim shield_rate in nf_en; the dim of unit_type_bits is 0. The dimension of nf_al = 4+0+0 = 4, the dimension of nf_en = 3+1+0 = 4. The concrete state form is [agent_0_health_rate, agent_0_weapon_cooldown_rate, agent_0_relative_x_to_map_center, agent_0_relative_y_to_map_center, agent_1_health_rate, agent_1_weapon_cooldown_rate, agent_1_relative_x_to_map_center, agent_1_relative_y_to_map_center, enemy_0_health_rate, enemy_0_relative_x_to_map_center, enemy_0_relative_y_to_map_center, enemy_0_shield_rate, last_actions]. Key states should focus on the health and shield status of the enemy and the health of our agents.,
    Key_states: {
      init:all agents and enemies are alive,
      key_state_1: enemy_0's shield is significantly damaged,
      key_state_2: enemy_0 is significantly damaged,
      key_state_3: enemy_0 is about to die,
      success:all enemies are dead and at least one agent is alive
      },
    Discriminators: {
      key_state_1: def iskeystate1(state): return 1 if state[11] <0.6 else 0,
      key_state_2: def iskeystate2(state): return 1 if state[8] <0.6 else 0,
      key_state_3: def iskeystate3(state): return 1 if state[8] <0.05 else 0,
      }

```
      Subspace_states: {
         key_state_1: [11],
         key_state_2: [8],
         key_state_3: [8],
         }
   }
```
**USER:**

**(LLM Rethinking)**
Please carefully check your response to make sure it meets the task requirements and notes without including unnecessary details. Also, confirm that the discriminator functions do not use any undefined variables.

**ASSISTANT: (LLM response)**

**......** (We have omitted the intermediate LLM outputs to maintain conciseness, retaining only the initial generation, as the differences between the initial and rechecked generations are minimal in the absence of errors.)

---

MPE Prompt and Response Example

**USER:**

**(Task_Description)**
Two agents are in a 30x30 room with the coordinate of the point in the upper left corner is (0,0) and the point in the lower right corner is (29,29), separated by a wall with a door.
The door can not be directly open. There is a hidden switch in each room (The switch coordinates are unknown and differ from the coordinate of the door.). The door will be open only when one of the switch is occupied by an agent otherwise close. Both agents begin in the left half of the room(agent0.x<15 and agent1.x<15).
The task is considered successfully completed as long as both agents are located in the right half-room(agent0.x>15 and agent1.x>15).

**(State_Form)**
In each step we can get current state. The state is a 1-dim list:
(agent0.x, agent0.y, agent1.x, agent1.y,door.open).

**(Role_Instruction)(Template)**
Your role is to give several critical key states in the task which we should try to reach and generate the corresponding discriminator function for each key state which can discriminate if the input state has reached the key state.
Note:
   1. Don't use the information you are not told.
   2. The code should be as generic as possible.
   3. The discriminator functions for different key states should be independent.
   4. Try not to put too much distance between the key states.
Please think step by step and adhere to the following JSON format (just replace the () with your answer):
{
   Thought: (Please fully understand the task. Think step by step and give your thought about the concrete state here),
   Key states: {
      init: init state,
      key_state_1: (description),
      key_state_2: (description),
```

        ...,
        success: success state
    },

Discriminators: {
        key_state_1: (a python function with the form of def iskeystate1(state): ... return 0 or 1),
        key_state_2: (a python function with the form of def iskeystate2(state): ... return 0 or 1),
        ...
    },
    Subspace_states: {
        key_state_1: (extract subspace state used in corresponding discriminator functions, such as [1] or [2,5]),
        key_state_2: (extract subspace state used in corresponding discriminator functions, such as [1] or [2,5]),
        ...
    }
}

**ASSISTANT: (LLM response)**

{
    Thought: To complete the task, we need to find the hidden switch in one of the rooms to open the door, then move both agents through the door to the right half of the room. We should establish key states that represent the discovery of the switch, the opening of the door, and the successful transition of both agents to the right half of the room. We'll assume that agent0 is responsible for finding the switch and agent1 is responsible for moving through the door once it's open.,
    Key_states: {
        init:Initial state where both agents are in the left half of the room.,
        key_state_1:Agent0 has found the switch and is standing on it, opening the door,
        key_state_2:Agent1 has moved through the open door to the right half of the room while agent0 is still on the switch,
        key_state_3:Agent0 has joined agent1 in the right half of the room
    },
    Discriminators: {
        key_state_1:def iskeystate1(state): return 1 if state[4] == 1 and state[0] >15 else 0,
        key_state_2:def iskeystate2(state): return 1 if state[4] == 1 and state[2] >15 and state[0] <15 else 0,
        key_state_3:def iskeystate3(state): return 1 if state[0] >15 and state[2] >15 else 0
    },
    Subspace_states: {
        key_state_1: [4,0],
        key_state_2: [0,2,4],
        key_state_3: [0,2],
    }
}

**USER:**

**(LLM Rethinking)**
Please carefully check your response to make sure it meets the task requirements and notes without including unnecessary details. Also, confirm that the discriminator functions do not use any undefined variables.

> **ASSISTANT: (LLM response)**
>
> **......** (We have omitted the intermediate LLM outputs to maintain conciseness, retaining only the initial generation, as the differences between the initial and rechecked generations are minimal in the absence of errors.)

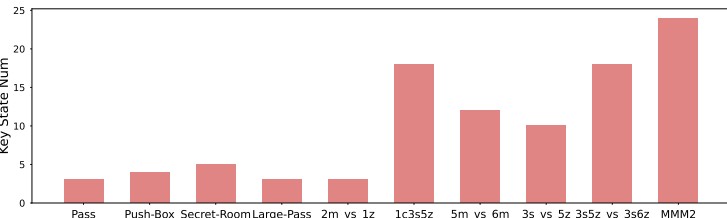

Figure 9: The number of LLM-generated key state discriminator functions.

Notably, the number of key states is primarily determined by LLM. For each task, we only prompt LLM to prevent it from generating too few functions according to the complexity of the environment. Specifically, as detailed in Appendix D, we instruct LLM to generate several critical key states for MPE and more than 2*n_enemies critical key states for SMAC.

As shown in Fig. 9, we summarize the number of LLM-generated key state discriminator functions. It is notable that the number of discriminator functions increases with the difficulty of the task or the number of interactive objects in the environment, which aligns with intuition.

Additionally, we have omitted the intermediate LLM outputs to maintain conciseness in this section, retaining only the initial generation, as the differences between the initial and rechecked generations are minimal in the absence of errors.

# E  IMPLEMENTATION DETAILS

## E.1  LEMAE AND BASELINES

**LEMAE**: Our code is based on the widely-used code framework pymarl2 at `https://github.com/hijkzzz/pymarl2`. In this study, we have integrated our method with several base algorithms **IPPO**, **QMIX**, **QPLEX**, and **VMIX**. Throughout the integration process, we have refrained from modifying the algorithmic implementation and have maintained consistency in parameters, including batch size, learning rate, and loss coefficients, in alignment with the configurations of the base algorithms.

**EITI** and **EDTI** (Wang et al., 2019): We compare our method with **EITI** and **EDTI** on MPE tasks proposed in Liu et al. (2021). We use the experiment results reported in Liu et al. (2021), which found that these algorithms perform poor because a long rollout (512 steps × 32 processes) between model updates is used.

**CMAE** (Liu et al., 2021): We compare our method with **CMAE** on MPE and SMAC tasks. On MPE tasks, the results of CMAE are reproduced using the publicly available code released by the authors at `https://github.com/IouJenLiu/CMAE`. As CMAE lacks an implementation for SMAC, we use the results reported in the original paper.

**MAVEN** (Mahajan et al., 2019): We use the code at `https://github.com/starry-sky6688/MARL-Algorithms`, which contains pytorch implementations of various MARL algorithms on SMAC, like the choice in LAIES (Liu et al., 2023).

**RODE** (Wang et al., 2020b) and **MASER** (Jeon et al., 2022): We utilized the publicly available code provided by the authors, accessible at `https://github.com/Jiwonjeon9603/MASER` and `https://github.com/TonghanWang/RODE`, respectively. Default configurations were employed, and their suboptimal performance is also documented in LAIES (Liu et al., 2023).

**LAIES** (Liu et al., 2023): We employed the publicly accessible code provided by the authors, which can be accessed at `https://github.com/liuboyin/LAIES`. When conducting experiments on SMAC, we adhered to the default configurations and external states. Notably, the original LAIES paper evaluation did not include assessments on the MPE. Consequently, we integrated the MPE environment into the LAIES codebase, designating the external states to represent the door status or the position of the box.

**ELLM** (Du et al., 2023): Since the tasks in this work have clearly defined goals, we minimize LLM inference costs by following the ELLM methodology but adapting its goal generation to occur only once at the start of the training. Consistent with the hyperparameters in the official codebase `https://github.com/yuqingd/ellm`, we set the similarity threshold to 0.99, rewarding only when the goal is achieved. We rely on LLM-generated functions to verify goal achievement, which we found to be more effective than directly using semantic similarity-based rewards.

For all algorithms, we ensure the same environmental settings, including observation space, environment reward function, and so on.

### E.2   COMPARISON WITH LLM REWARD DESIGN

We conduct additional experiments comparing LEMAE with a baseline called **Eureka-si**, which can be seen as a single-iteration variant of Eureka (Ma et al., 2023), where LLM designs rewards directly. For fairness, we does not adopt evolutionary optimization in Eureka and use LLM to generate reward functions with the same role instructions as in Eureka, while maintaining designs like Self-Check as in LEMAE. As shown in Fig. 3, Eureka-si is comparable to LEMAE in simple tasks like Push-Box but fails in challenging tasks with characteristics like partial observability, such as Pass, where hidden switches make it difficult to design effective reward functions. In contrast, LEMAE consistently demonstrates impressive performance. Notably, comparing LEMAE with Eureka directly would be unfair since Eureka's evolutionary search requires multiple training iterations and candidates, leading to significantly more sampling and training than LEMAE. Overall, LEMAE's advantage over RL algorithms lies in incorporating prior knowledge from the LLM, and its advantage over other LLM-based methods is due to our designs for better LLM incorporation, such as utilizing discrimination, SHIR, and KSMT.

### E.3   CONNECTION AND COMPARISON WITH HER

The proposed Key State-Guided Exploration is similar to Hindsight Experience Replay (**HER**) (Andrychowicz et al., 2017) in form, where key states and subgoals are certain states from sampled trajectories. However, unlike HER, which samples goals from memory using random or heuristic strategies and often struggles with shaped rewards, our method incorporates LLM priors for more targeted goal selection (key states localization). Additionally, the proposed KSMT and SHIR facilitate organized exploration and enhanced reward guidance.

We conduct additional experiments to further confirm the advantages of our method. We evaluate HER with IPPO as the backbone in MPE. We use the future strategy for goal selection, as proposed in the HER paper, and employ a reward function based on the Manhattan Distance, which we find to be the best match. However, as depicted in Fig. 3, HER does not perform well on both MPE tasks. This outcome suggests that the random sampling strategy for goals may not be sufficient, underscoring the importance of incorporating LLM priors for efficient exploration as we proposed.

### E.4   TASKS

### E.4.1   MULTIPLE-PARTICLE ENVIRONMENT (MPE)

The Multiple-Particle Environment serves as a widely-adopted benchmark for multi-agent scenarios. In this work, we employ tasks specifically crafted for evaluating multi-agent exploration, proposed by Wang et al. (2019). The implementation utilized in this study is based on the work by Liu et al. (2021). In this section, we provide details of the four sparse-reward tasks we adopted.

- *Pass*: In the *Pass* task, depicted in Fig. 10a, two agents are positioned in a room of 30 x 30 grid. The room is divided into two halves by a wall featuring a door. Each half-room contains an

invisible switch, the details of which are not contained in the state or prompt for LLM. The door permits passage only when one of the switches is occupied by an agent. Initially situated within the left half-room, both agents must cooperate to transfer to the right half-room. The external reward function is denoted as $r_E = I(two\ agents\ are\ in\ the\ right\ room)$, where $I$ represents the indicator function.

● *Secret-Room*: *Secret-Room* is an extension task of *Pass*. As illustrated in Fig. 10b, the configuration comprises one sizable room on the left and three smaller rooms on the right, interconnected by three doors. Within each room, there is an invisible switch; notably, the switch in the left room has the capability to control all three doors, whereas each right room's switch exclusively controls its respective door. The grid size is 25 x 25. Two agents are initialized within the left room and are required to collaborate in order to transition to the real target room, which is the right room 2. The external reward function is denoted as $r_E = I(two\ agents\ are\ in\ the\ right\ room\ 2)$, where $I$ represents the indicator function.

● *Push-Box*: As depicted in Fig. 10(c), two agents and a box are initially positioned within a 15 x 15 grid. To successfully move the box, both agents must simultaneously exert force in the same direction. The task is deemed accomplished when the box is successfully pushed to the wall. The external reward function is denoted as $r_E = I(the\ box\ is\ pushed\ to\ the\ wall)$, where $I$ represents the indicator function.

● *Large-Pass*: *Large-Pass* is a direct extension task of *Pass* by enlarging the grid dimensions to 50 x 50, which makes it more challenging. The external reward function aligns with that of the *Pass* task.

The details of these tasks, including observation space and action space, are listed in Table 3.

Table 3: Details of MPE tasks

| MPE tasks | n_agents | observation space | state space | action space |
|---|---|---|---|---|
| Pass | 2 | 5 | 5 | 4 |
| Secret-Room | 2 | 5 | 5 | 4 |
| Push-Box | 2 | 6 | 6 | 4 |
| Large-Pass | 2 | 5 | 5 | 4 |

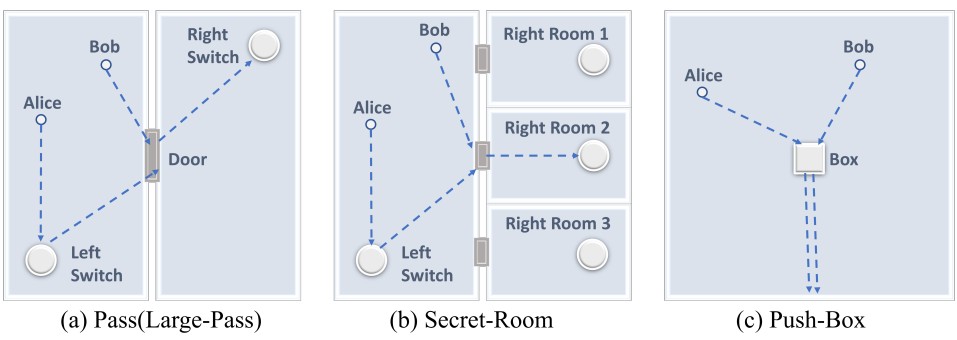

(a) Pass(Large-Pass)   (b) Secret-Room   (c) Push-Box

Figure 10: MPE Tasks.

### E.4.2 STARCRAFTII MULTI-AGENT CHALLENGE (SMAC)

StarCraftII Multi-Agent Challenge (SMAC) (Samvelyan et al., 2019a) is a widely-used benchmark in the realm of cooperative multi-agent reinforcement learning research (Rashid et al., 2018; Shao et al., 2023; Liu et al., 2023; Shao et al., 2024). Derived from the renowned real-time strategy game StarCraft II, SMAC concentrates specifically on decentralized micromanagement scenarios rather than the full game. Typically, the tasks within SMAC adopt a dense-reward framework, wherein agents receive dense rewards for damage received, attacking and eliminating enemies. To promote the need for exploration, we adopt fully sparse-reward versions of tasks in SMAC where agents are solely rewarded upon the successful elimination of all enemies. The external reward function is denoted

as $r_E = I(all\ enemies\ are\ eliminated)$, where $I$ represents the indicator function. Notably, this sparse-reward setting differs from the sparse SMAC, which can be called semi-sparse SMAC, used in some previous studies (Jeon et al., 2022; Jo et al., 2023), where agents are rewarded when one or all enemies die or when one ally dies. In addition, to validate the versatility of LEMAE across diverse scenarios, we conducted experiments on six maps with different difficulty and diverse agent numbers, as illustrated in Table 4 and Table 5. We use the version of SC2.4.10. Please refer to the official document[1] for more details.

Table 4: SMAC tasks.

| Task | Ally Units | Enemy Units | Type | Difficulty |
|---|---|---|---|---|
| 1c3s5z | 1 Colossi, 3 Stalkers, 5 Zealots | 1 Colossi, 3 Stalkers, 5 Zealots | heterogeneous & symmetric | Easy |
| 2m_vs_1z | 2 Marines | 1 Zealot | micro-trick: alternating fire | Easy |
| 3s_vs_5z | 3 Stalkers | 5 Zealots | micro-trick: kiting | Hard |
| 5m_vs_6m | 5 Marines | 6 Marines | homogeneous & asymmetric | Hard |
| 3s5z_vs_3s6z | 3 Stalkers, 5 Zealots | 3 Stalkers, 6 Zealots | heterogeneous & asymmetric | Super-Hard |
| MMM2 | 1 Medivac, 2 Marauders, 7 Marines | 1 Medivac, 3 Marauders, 8 Marines | heterogeneous & asymmetric | Super-Hard |

Table 5: Details of SMAC tasks

| SMAC tasks | n_agents | n_enemies | observation space | state space | action space |
|---|---|---|---|---|---|
| 2m_vs_1z | 2 | 1 | 16 | 26 | 7 |
| 1c3s5z | 9 | 9 | 162 | 270 | 15 |
| 3s_vs_5z | 3 | 5 | 48 | 68 | 11 |
| 5m_vs_6m | 5 | 6 | 55 | 98 | 12 |
| 3s5z_vs_3s6z | 8 | 9 | 136 | 230 | 15 |
| MMM2 | 10 | 12 | 176 | 322 | 18 |

### E.4.3 A Brand New Task: *River*

To exclude the probability that LEMAE's success relies on LLM's familiarity with the chosen tasks, we've designed a brand new task, termed *River*, which LLM has never encountered before. The task is detailed as follows:

The *River* task is adapted from the Multiple-Particle Environment and its map is illustrated in Fig. 8a. Two agents, Alice and Bob, are placed in a 30 x 30 grid field intersected by two rivers running vertically and horizontally. A mountain in the bottom-left corner obstructs the passage. Alice and Bob start randomly in the top-left part of the field and need to move to the bottom-right part. However, Alice is afraid of water and cannot cross the river unless Bob stays in the river to act as a bridge for her.

The observation space is discrete with four dimensions, representing the positions of two agents, i.e., $o = [x_1, y_1, x_2, y_2]$. The action space is also discrete, allowing movement in four directions. Agents receive a positive reward only when both agents reach the bottom-right corner of the field.

### E.5 Hyperparameters

In LEMAE, we introduce three important hyperparameters: extrinsic reward scaling rate $\alpha$, intrinsic reward scaling rate $\beta$, and high randomness epsilon $\epsilon_h$. Notably, the low randomness epsilon $\epsilon_l$ is the hyperparameter in the base algorithms, such as 0.05 for QMIX and 0.0 for IPPO.

---

[1] https://github.com/oxwhirl/smac/blob/master/docs/smac.md

For MPE, we adopt $\{\alpha = 10, \beta = 0.1, \epsilon_h = 1\}$ on *Pass*, *Secret-Room*, and *Large-Pass* and use $\{\alpha = 10, \beta = 0.05, \epsilon_h = 0.2\}$ on *Push-Box*.

For SMAC, we adopt $\{\alpha = 50, \beta = 1, \epsilon_h = 0.5\}$ on *MMM2* and *1c3s5z*, $\{\alpha = 10, \beta = 1, \epsilon_h = 0.5\}$ on *3s_vs_5z* and *2m_vs_1z*, $\{\alpha = 1, \beta = 1, \epsilon_h = 0.5\}$ on *5m_vs_6m* and *3s5z_vs_3s6z*.

### E.6 RESOURCES

We use a server with 8*NVIDIA RTX 3090 GPUs, and 2*AMD 7H12 CPUs to run all the experiments. Without specifying, each setting is repeated for 5 seeds. For one seed in SC2, the running time ranges from approximately 50 minutes to 12 hours. For MPE, the running time varies from around 3 to 7 hours. The input for each LLM (GPT-4-1106-preview) inference comprises approximately 600-4000 tokens (0.006-0.04 dollars), yielding an output of about 300-1600 tokens (0.009-0.048 dollars).

## F ADDITIONAL EXPERIMENTAL RESULTS

### F.1 MORE COMPLICATED BENCHMARK: SMACV2

We have evaluated LEMAE on three typical tasks, protoss_5_vs_5, terran_5_vs_5, and zerg_5_vs_5, in SMACv2 (Ellis et al., 2024) to demonstrate its effectiveness under stochastic dynamics settings. We utilized the typical hyperparameters for both LEMAE and LAIES as used in SMAC. Additionally, we evaluate several typical baselines, including a generic intrinsic reward method, E3B Henaff et al. (2022), and two LLM-based methods, ELLM Du et al. (2023) and ProgressCount Sarukkai et al. (2024). Notably, all these additional baselines were originally proposed for single-agent scenarios, and we made proper adaptations for multi-agent settings to ensure a fair comparison. As shown in Fig. 11, LEMAE achieves outstanding performance, confirming its applicability to such settings and the effectiveness of its specific design choices for MARL. This result further demonstrates LEMAE's potential for real-world scenarios with complexity and stochasticity.

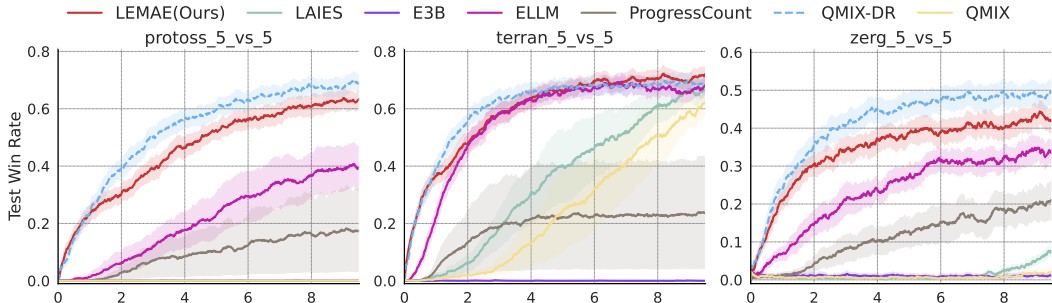

Figure 11: Evaluating LEMAE against baseline methods across three maps in SMACv2, which features greater randomness, using the evaluation metric of test win rate. **QMIX-DR** is QMIX with dense rewards in the original SMACv2, serving as an upper bound, with its intrinsic rewards designed by a human expert. This result further demonstrates LEMAE's potential for real-world scenarios with complexity and stochasticity.

### F.2 EXTENDING LEMAE BEYOND SYMBOLIC TASKS

This work primarily focuses on tasks with symbolic state spaces, where states are represented as symbolic arrays describing the agent and environment. As discussed in Appendix B.3, to extend LEMAE from symbolic tasks to vision-based tasks, we can exchange the LLM for a multi-modal LM in LEMAE for key state localization. To confirm the applicability of LEMAE to vision-based tasks, we conduct a demonstrative experiment: We extend the task *Pass* to a vision-based task *Visual-Pass*, as illustrated in Fig. 12a. We prompt a LLM to define key states with the same task description and role instruction as proposed in Sec. 4.2 and use the LLM-generated definition as the prompt for a **Vision Language Model (GPT-4o)**. Then, it is prompted to discriminate key states in the randomly sampled states. GPT-4o achieves a **98%** accuracy rate in discriminating key states among

the 50 sampled image states. This confirms that with a proper extension of the LLM, LEMAE can eliminate dependence on state semantics and be applied to other tasks such as visual-input. As shown in Fig. 12c, we evaluate the effectiveness of LEMAE on the image-based task *Visual-Pass* by using a VLM as the discriminator and object pixel detection for reward calculation. The experimental results show LEMAE's broader applicability.

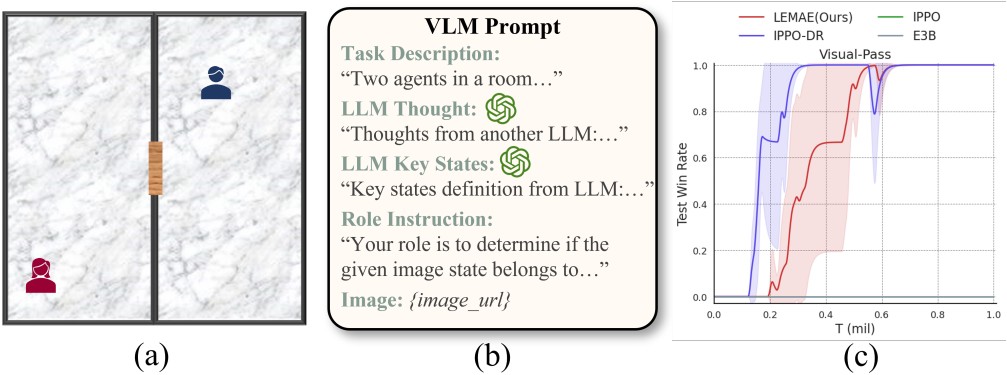

(a)         (b)         (c)

Figure 12: (a) An illustration of the vision-based task *Visual-Pass*. (b) The prompt for the Vision Language Model (VLM), which includes the task description, role instruction, image state and key states definition provided by another LLM. The VLM is tasked with determining whether the given image state corresponds to a key state. (c) Training curves of LEMAE and IPPO on the image-based *Visual-Pass* tasks, where **IPPO-DR refers to IPPO trained with human-designed dense rewards.**

### F.3 ABLATION STUDIES ON MIXED-RANDOMNESS EXPLORATION

As demonstrated in Fig 13, we conduct an ablation study on mixed-randomness exploration within the *3s_vs_5z* map. Results indicate that LEMAE exhibits insensitivity to the parameter $\epsilon_h$, provided that the level of randomness remains moderate, as opposed to being excessively extreme (0.1 or 0.9). Besides, the effectiveness of our design is highlighted through a comparison between LEMAE and its variants, namely, only leaf node and LEMAE w/o KSMTE.

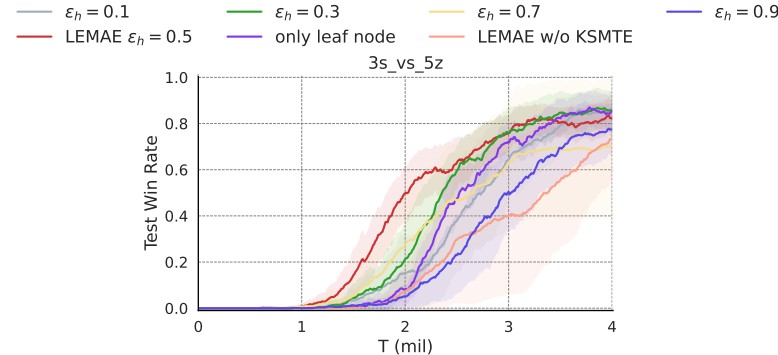

Figure 13: Ablation study on mixed-randomness exploration. The default hyperparameter used in LEMAE is $\epsilon_h = 0.5$. Robustness evaluation included variations in $\epsilon_h = [0.1, 0.3, 0.7, 0.9]$. The only leaf node denotes simply using $\epsilon_l$ for the non-leaf node and $\epsilon_h$ for the leaf node. The LEMAE w/o KSMTE denotes LEMAE without mixed-randomness exploration.

### F.4 EXPERIMENTS FOR SINGLE-AGENT SETUPS

Indeed, we propose LEMAE as a general approach for LLM-empowered efficient exploration in reinforcement learning, applicable to both single-agent and multi-agent settings. We underscore the evaluation of its performance in multi-agent settings due to its inherent complexity.

As the proposed method can seamlessly extend to single-agent scenarios, we introduce a single-agent variant of MPE and assess PPO (Schulman et al., 2017) and PPO-based LEMAE across four tasks. We run each algorithm using three random seeds with 300k environment steps, using the evaluation metric of the test win rate. The following table shows that LEMAE can facilitate efficient exploration in single-agent scenarios.

Table 6: Final test win rate of LEMAE and PPO on single-agent variant of MPE tasks.

| Single MPE | PPO | LEMAE |
| --- | --- | --- |
| Single Pass | $0.00_{\pm 0.00}$ | $\mathbf{1.00}_{\pm \mathbf{0.00}}$ |
| Single Secret-Room | $0.00_{\pm 0.00}$ | $\mathbf{0.98}_{\pm \mathbf{0.01}}$ |
| Single Large-Pass | $0.00_{\pm 0.00}$ | $\mathbf{0.99}_{\pm \mathbf{0.01}}$ |
| Single Push-Box | $0.00_{\pm 0.00}$ | $\mathbf{0.96}_{\pm \mathbf{0.08}}$ |

## F.5 DISCUSSION ON THE KSMT

Using KSMT could pose a limitation due to potential memory costs in certain scenarios. However, this has not been a significant issue in our experiments, as the key states are relatively few, primarily focusing on the most critical ones, with a natural sequential relationship typically existing between them. Notably, LEMAE is also compatible with other memory structures, such as Directed Acyclic Graphs (DAGs), which could be an interesting direction for future exploration.

To demonstrate the effectiveness of LEMAE with other memory structures, in scenarios where task completion follows a linear pattern (e.g., $Init \rightarrow A \rightarrow B \rightarrow Success$), we employ a more efficient strategy by using a KSMT variant with a single branch representing the sequential order of key states. Specifically, we systematically assign a priority value to each key state, continuously updating it based on its occurrence order within the sequence of attained key states. The determination of the ranking of key states within the one-branch KSMT relies on this established priority.

As illustrated in Fig. 14, an ablation study is conducted to compare the performance between raw KSMT and the one-branch KSMT variant across six maps in SMAC. The results demonstrate the increased necessity of employing the one-branch KSMT variant for tasks involving a larger number of agents and greater complexity, such as *5m_vs_6m*, *3s5z_vs_3s6z*, and *MMM2*. Consequently, we have adopted the one-branch KSMT approach for these specific SMAC tasks: *5m_vs_6m*, *3s5z_vs_3s6z*, and *MMM2*.

### F.5.1 CONNECTIONS WITH GO-EXPLORE

Go-Explore (Ecoffet et al., 2019) is an influential work tackling exploration in RL. The similarities between our KSMT and the archive in Go-Explore lie in both methods organizing exploration through memory, i.e., by selecting possible historical states to explore. The differences and partial contributions of LEMAE are as follows: (1) Our key states are semantically meaningful and task-critical, whereas the archived states in Go-Explore are randomly explored; (2) Our KSMT samples key states based on actual key states transitions, enhancing its reliability; (3) We propose Explore with KSMT to balance exploration and exploitation, thereby reducing exploration complexity by focusing on a more meaningful state subspace.

## F.6 WORKING WITH DENSE REWARD SETTINGS

We also evaluate LEMAE in tasks with dense rewards in SMAC, denoted as LEMAE-DR. As shown in Fig. 15, the results confirm that LEMAE-DR facilitates efficient exploration in **both dense and sparse reward** settings, highlighting the main contribution of our method. Additionally, LEMAE-DR achieves better convergence than LEMAE due to the guidance provided by dense rewards.

## F.7 APPLICATIONS IN ROBOTICS CONTROL

We evaluate LEMAE on MaMuJoCo (Peng et al., 2021), a MARL robotics benchmark. We adapt the tasks to emphasize exploration with sparse rewards for achieving high velocity $v$, where the

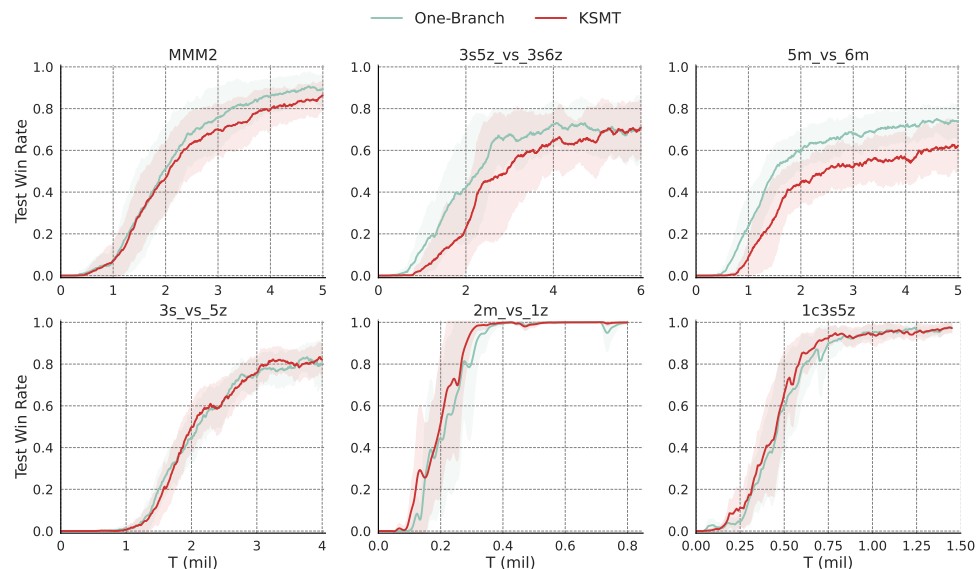

Figure 14: Ablation study conducted to compare the performance between raw KSMT and the one-branch KSMT variant across six maps in SMAC.

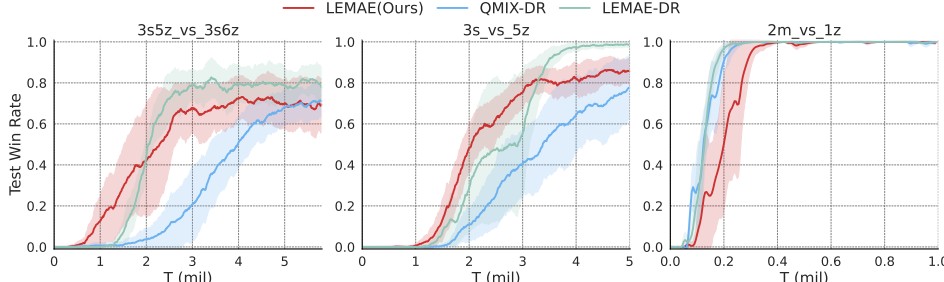

Figure 15: Evaluating LEMAE with dense rewards across three maps in SMAC using the evaluation metric of test win rate. **LEMAE-DR** is LEMAE with dense rewards in the original SMAC, which effectively ensures efficient exploration and achieves better convergence.

reward function is defined as $I(v > v_{threshold})$. We use HAPPO (Kuba et al., 2021) as the RL backbone. As shown in Fig. 16, LEMAE achieves performance comparable to the baseline trained with human-designed dense rewards. This observation is consistent with previous conclusions. LEMAE benefits from the reliability of the proposed LLM key state discrimination, and the results underscore LEMAE's potential for handling complex tasks.

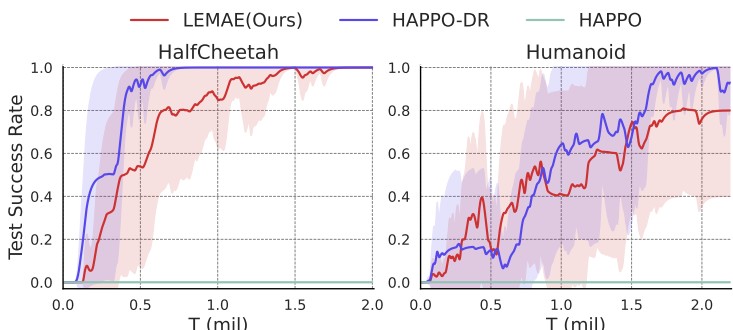

Figure 16: Evaluating LEMAE on two tasks in MaMuJoCo using the metric of test success rate. **HAPPO-DR** is HAPPO with dense rewards in the original tasks.

## F.8 ADDITIONAL BASELINES

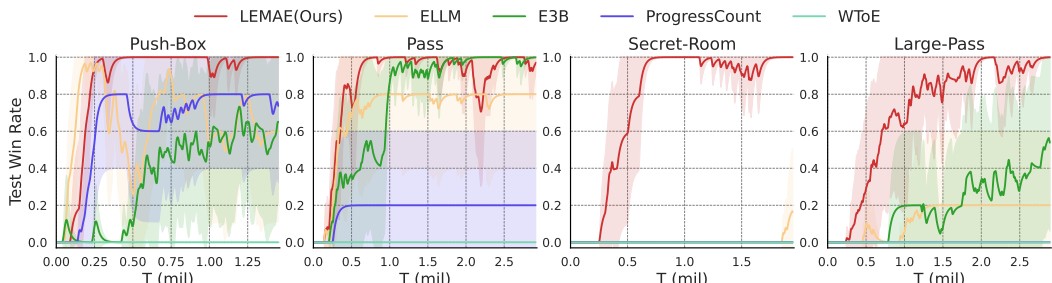

Figure 17: Evaluating LEMAE and additional baselines on tasks in MPE.

As shown in Fig. 17, we compare LEMAE with three additional baselines:

- ProgressCounts (Sarukkai et al., 2024) is a recent LLM-based method which combines LLM reward design and count-based exploration. Still, LEMAE consistently outperformed ProgressCounts, suggesting the importance of our designs—specifically, the use of discrimination, SHIR, and KSMT—in more effectively integrating LLMs compared to other LLM-based methods.

- WToE (Dong et al., 2023) focuses on when to explore by identifying discrepancies between the actor policy and a short-term inferred policy that adapts to environmental changes. WToE does not utilize intrinsic reward as guidance, which may make it unsuitable for tackling long-horizon sparse-reward tasks.

- E3B (Henaff et al., 2022) is a generic intrinsic reward method. The results witness LEMAE's superior exploration efficiency, implying the effectiveness of key states.

These comparisons highlight the effectiveness of LEMAE's specific design choices in incorporating LLMs and applying them to MARL.

## F.9 ADDITIONAL RESULTS

In this section, we provide some additional experimental results.

Table 7: Final test win rate of LEMAE and comparable baseline (CMAE) on MPE tasks.

| MPE | LEMAE (Ours) | CMAE |
|---|---|---|
| Pass | $\mathbf{1.00}_{\pm\mathbf{0.00}}$ | $0.75_{\pm 0.43}$ |
| Secret-Room | $\mathbf{1.00}_{\pm\mathbf{0.00}}$ | $0.80_{\pm 0.40}$ |
| Push-Box | $\mathbf{1.00}_{\pm\mathbf{0.00}}$ | $0.80_{\pm 0.40}$ |
| Large-Pass | $\mathbf{1.00}_{\pm\mathbf{0.00}}$ | $0.00_{\pm 0.00}$ |

Table 8: Final test win rate of LEMAE and comparable baseline (LAIES) on SMAC tasks. QMIX-DR denotes training QMIX with dense reward.

| SMAC | LEMAE (Ours) | LAIES | QMIX-DR |
|---|---|---|---|
| 1c3s5z | $\mathbf{0.98}_{\pm\mathbf{0.02}}$ | $0.89_{\pm 0.09}$ | $\mathbf{0.99}_{\pm\mathbf{0.01}}$ |
| 2m_vs_1z | $\mathbf{1.00}_{\pm\mathbf{0.01}}$ | $0.73_{\pm 0.24}$ | $\mathbf{1.00}_{\pm\mathbf{0.01}}$ |
| 3s_vs_5z | $\mathbf{0.83}_{\pm\mathbf{0.07}}$ | $0.10_{\pm 0.12}$ | $0.66_{\pm 0.16}$ |
| 5m_vs_6m | $0.74_{\pm 0.08}$ | $0.68_{\pm 0.10}$ | $\mathbf{0.78}_{\pm\mathbf{0.08}}$ |
| 3s5z_vs_3s6z | $\mathbf{0.73}_{\pm\mathbf{0.14}}$ | $0.45_{\pm 0.35}$ | $\mathbf{0.73}_{\pm\mathbf{0.07}}$ |
| MMM2 | $\mathbf{0.89}_{\pm\mathbf{0.08}}$ | $0.62_{\pm 0.25}$ | $\mathbf{0.90}_{\pm\mathbf{0.05}}$ |

Table 9: Comparing LEMAE with SOTA baseline CMAE across four maps in MPE using the metric of the number of exploration steps (in thousand) taken to find the success state

| MPE | LEMAE (Ours) | CMAE | Acceleration rate |
|---|---|---|---|
| Pass | $\mathbf{153.1}_{\pm\mathbf{20.7}}$ | $2114.8_{\pm 157.4}$ | **13.8** |
| Secret-Room | $\mathbf{316.6}_{\pm\mathbf{134.6}}$ | $1448.5_{\pm 467.2}$ | **4.6** |
| Push-Box | $\mathbf{159.0}_{\pm\mathbf{42.5}}$ | $972.3_{\pm 887.3}$ | **6.1** |
| Large-Pass | $\mathbf{446.9}_{\pm\mathbf{256}}$ | $>3000$ | $\mathbf{>6.7}$ |

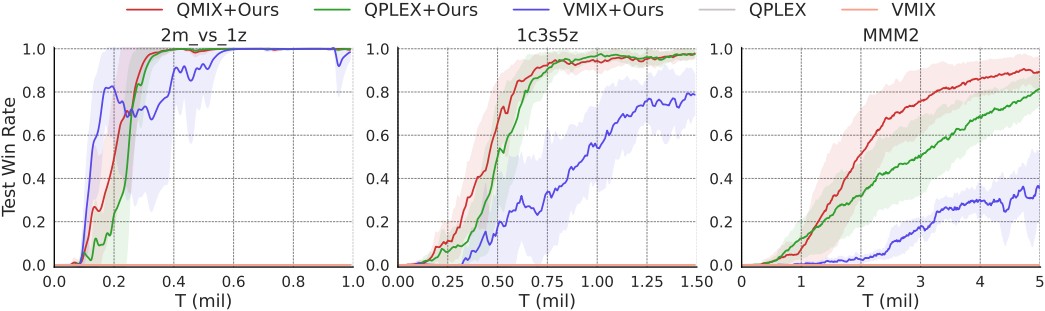

Figure 18: The training curve while evaluating the efficacy of combining our method with various algorithms.

As demonstrated in Table 7 and Table 8, we augment the final test win rate of our proposed method, LEMAE, with comparable baseline algorithms in MPE and SMAC tasks. This augmentation serves to elucidate the superior performance of our method. It is pertinent to note that baseline algorithms, the performance of which has been demonstrated to be poor in the training curves, are omitted from the tables for conciseness.

As demonstrated in Table 9, we compare LEMAE with SOTA baseline CMAE using the metric of the number of exploration steps taken to find the success state. The results indicate a significant exploration acceleration rate, up to 10x, underscoring LEMAE's superior efficiency.

Moreover, as illustrated in Fig. 18, we supplement the training curve while evaluating the efficacy of combining our method with various algorithms, i.e., QPLEX and VMIX.

