# OpenReview forum: "Choices are More Important than Efforts: LLM Enables Efficient Multi-Agent Exploration"
_ICLR.cc/2025/Conference — Submitted to ICLR 2025_

### Official Review · Reviewer_5ctA · 2024-10-18

**Soundness:** 1
**Presentation:** 1
**Contribution:** 1
**Rating:** 1
**Confidence:** 5

**Summary:**

This paper presents an approach for using LLMs to identify key states for multi-agent reinforcement learning agents to more efficiently explore in procedurally generated environments. Given a trajectory, the LLM generates key states, which are then organized in a key states memory tree, and used in a hindsight experience replay-like approach.

**Strengths:**

- I believe the paper provides sufficient detail in the appendix to reproduce it.
- The direction of the paper, addressing hard exploration in multi-agent systems, is an important one.

**Weaknesses:**

## Major
A) The main results of the paper center around The StarCraft Multi-Agent Challenge (SMACv1) by Samvelyan et al. (2019). However, it is well established by now that SMACv1 is saturated and too simple. The authors should use the 2022 released SMACv2 [1] for their experiments. From the SMACv2 paper: “after years of sustained improvement on SMAC, algorithms now achieve near-perfect performance. In this work, we conduct new analysis demonstrating that SMAC lacks the stochasticity and partial observability to require complex *closed-loop* policies. In particular, we show that an *open-loop* policy conditioned only on the timestep can achieve non-trivial win rates for many SMAC scenarios.”

B) Using language observations or LLMs to guide exploration of RL agents for hard exploration problems has been extensively studied in the single-agent setup. The authors have not covered important very related methods (e.g. [2-4]) in this space. In particular I would expect Motif [3] and OMNI [4] to be tried out as a baseline in the multi-agent RL (MARL) setting. In fact, I would go as far and say that the proposed method is not MARL-specific at all, so the authors really need to show that it provides benefits over existing exploration methods in the single agent setting.

C) Related to B, it is even unclear to me whether explicit key states are necessary for exploration in the MARL environments used here. I would like to see a comparison to MARL with general intrinsic reward mechanisms such as E3B [5] to answer the question whether key states significantly improve exploration over generic intrinsic reward methods.

D) The authors claim that “KSMT helps avoid redundant exploration throughout the state space, particularly beneficial in more complicated real-world scenarios” despite that no experiments on such real-world problems are provided. Either tone it down or provide additional empirical evidence in more-real world environments. I would be very interested in the latter, in particular seeing experiments on environments that do not have a symbolic observation space.

[1] Ellis, B., Moalla, S., Samvelyan, M., Sun, M., Mahajan, A., Foerster, J. N., & Whiteson, S. (2022). SMACv2: An Improved Benchmark for Cooperative Multi-Agent Reinforcement Learning. arXiv. https://doi.org/10.48550/arXiv.2212.07489

[2] Mu, J., Zhong, V., Raileanu, R., Jiang, M., Goodman, N., Rocktäschel, T., & Grefenstette, E. (2022). Improving Intrinsic Exploration with Language Abstractions. arXiv. https://doi.org/10.48550/arXiv.2202.08938

[3] Klissarov, M., D’Oro, P., Sodhani, S., Raileanu, R., Bacon, P.-L., Vincent, P., … Henaff, M. (2023). Motif: Intrinsic Motivation from Artificial Intelligence Feedback. arXiv. https://doi.org/10.48550/arXiv.2310.00166

[4] Zhang, J., Lehman, J., Stanley, K., & Clune, J. (2023). OMNI: Open-endedness via Models of human Notions of Interestingness. ICLR 2024.

[5] Henaff, M., Raileanu, R., Jiang, M., & Rocktäschel, T. (2023). Exploration via Elliptical Episodic Bonuses. arXiv. https://doi.org/10.48550/arXiv.2210.05805


## Minor
- LLMs instead of LLM (e.g. in the title); also "Efforts" -> "Effort" in the title
- Be more precise with what you mean by “redundant exploration”
- Use more formal language (e.g. not “brand-new” in line 089)
- L160: “to clarity” -> “to clarify”
- Figure 2 is somewhat unclear to me, in particular the LLM-mapping from Prompt to Response. I particularly find it hard to map what’s formalized in 4.2 (e.g. L209 to L212) to what’s shown in Figure 2. Also, what feedback flows directly back from response to prompt?

**Questions:**

- L190: What do you mean by “the initial policy is asymmetric”?
- L247: What’s the rationale for using Manhattan Distance instead of, for example, cosine similarity?

What the authors would have to demonstrate to see an improved rating from me:
- Redo experiments on SMACv2 (A)
- Add baselines Motif and OMNI, requiring them to be evaluated on MARL (B)
- Add E3B baseline, again integrating that into the MARL setup, or, alternatively (since the proposed method is not MARL-specific) run the proposed method on single-agent procedurally generated environments. (C)
- Demonstrate that the proposed method also works for non-symbolic observations (D)

---

> ### Author Response · Authors · 2024-11-18
>
> _We sincerely appreciate Reviewer 5ctA's efforts and feedback. The following addresses the concerns._
>
> **1.SMACv2(A)**
>
> We clarify that (1) SMAC is a **standard benchmark** in MARL exploration studies; (2) **the fully sparse-reward SMAC remains far from saturated**; (3) we **have evaluated LEMAE on SMACv2 in Appendix F.1, revealing its potential for complex and stochastic scenarios**.
>
> **2.LLM-based methods(B)**
> - Compared to prior single-agent LLM-based methods, **LEMAE is a pioneering attempt to tackle the unique challenges of MARL with LLM for efficient exploration**: (1) LLM-based key states discrimination to provide reliable priors, particularly for **high-dimensional and partially observable MARL tasks**; (2) Labor-division-oriented prompt designs to **reduce manual effort and foster multi-agent cooperation**; (3) Subspace-based Hindsight Intrisic Reward to **improve RL guidance and alleviate manual reward design**; and (4) Key States Memory Tree to organize exploration with memory, **reducing complexity by focusing on a meaningful subspace within the exponentially expanded state space**.
> - We **have demonstrated LEMAE's superiority over the most relevant LLM-based methods in Fig.3**: LLM step-wise exploration goals generation, exemplified by ELLM[4]; LLM-based reward design, exemplified by Eureka[5]. Additionally, **LEMAE outperformed ProgressCounts[6], a recent method combining LLM reward design and count-based exploration**, in revised Appendix F.8.
> - **The suggested baselines are not suitable for our setup, as discussed in revised Sec.3**: Motif[1] requires pre-collected annotated datasets and extensive LLM inferences; OMNI[2] focuses on task sampling for skill acquisition in open-endedness environments, unlike our focus on single-task efficient exploration without a task pool.
>
> **3.Significance of Key States(C)**
>
> As shown in Fig.1, **key states play a crucial role in promoting efficient exploration in MARL**. They serve as meaningful intermediate states that guide task completion and exhibit a division of labor in our design, enhancing multi-agent collaboration.\
> As suggested, we **compare LEMAE with E3B**[3] in revised Appendix F.8. LEMAE's superior exploration efficiency implies **the effectiveness of key states over generic intrinsic reward methods**. We clarify that **the baselines also use distinct intrinsic reward mechanisms**, and LEMAE's advantage emphasizes the importance of key states.
>
> |Task|LEMAE|E3B|
> |-|-|-|
> |Pass|1.0±0.0|1.0±0.0|
> |Push-Box|1.0±0.0|0.7±0.4|
>
> **4.Application Scopes Clarification(D)**
>
> We clarify that **LEMAE primarily focuses on achieving efficient exploration in tasks with symbolic state, a common assumption in prior methods[5,6]**.\
> **LEMAE can also be extended to non-symbolic tasks. We have extended it to image-based tasks, *Visual-Pass*, in Appendix F.2**(LEMAE:1.0±0.0; IPPO(dense reward):1.0±0.0 ;IPPO:0.0±0.0)\
> We have refined the statement. KSMT reduces exploration complexity in tasks with complex symbolic state spaces by focusing on a meaningful state subspace(Fig.1), revealing its potential for real-world applications.
>
> **5.Methodology Explanations**
>
> We apologize for any confusion and have revised the manuscript for clarity. Below are the detailed explanations:
> - Redundant Exploration: refers to **exploration behaviors irrelevant to the corresponding tasks, as described in ELLM[4]**.
> - LLM generation: (1) LLM takes in the prompt template and task information and generates $m$(determined by LLM) key states' definitions and disciminator functions, which are easily extracted from the predefined JSON response through key-value matching. (2) Each discriminator function $\mathcal{F}_i$(**corresponding to the 'Iskeystate(s)' block in Fig.2**) takes in the state $s_t$ and outputs a boolean value to **tell whether $s_t$ is the key state $\kappa_i$, aligning with the annotation process from $s_t$ to $s_t^{\kappa_2}$ depicted in Fig.2**. (3) The **feedback, introduced in Line 226-232 (Self-Check), includes the prior response, rethinking queries, and execution errors**.
> - Asymmetric policy: As stated in Proposition 1, the initial policy is random, with a probability $p\in(0.5, 1)$ for moving right. This indicates **an asymmetry between right and left movements**.
> - Manhattan distance: is adopted due to its empirical effectiveness. **LEMAE is compatible with other distance metrics**, like cosine similarity, which may perform better in other applications.
>
> **6.Other Typos**
>
> We appreciate your suggestions regarding the writing typos and have revised the manuscript accordingly.
>
> [1]Motif: Intrinsic Motivation from Artificial Intelligence Feedback.\
> [2]OMNI: Open-endedness via Models of human Notions of Interestingness.\
> [3]Exploration via Elliptical Episodic Bonuses.\
> [4]Guiding pretraining in reinforcement learning with large language models\
> [5]Eureka: Human-level reward design via coding large language models\
> [6]Automated Rewards via LLM-Generated Progress Functions

---

> ### Comment · Reviewer_5ctA · 2024-11-18
>
> Thank you for your effort.
>
> A) I want to strongly but decisively push back here. Many of the authors of SMAC are also authors of SMACv2. I have confirmed with one of the authors of SMACv2 that since open-loop policies solve many of the problems in **SMAC, that benchmark should be seen as fundamentally broken by now and SMACv2 needs to be used instead**. I acknowledge the additional experiments on SMACv2 in the Appendix. However, the authors have tested only on the simplest SMACv2 tasks (first row of figure 6 in https://arxiv.org/abs/2212.07489) and, more importantly, even on these simplest SMACv2 tasks, **LEMAE performs worse than the baseline QMIX-DR that neither uses intrinsic rewards nor language abstractions or key states, thus putting in question the validity of the LEMAE method**. "LEMAE achieves outstanding performance" is a gross misrepresentation of how these results should be interpreted.
>
> B) I acknowledge the comment regarding OMNI and Motif, but I need to call out that the authors have completely ignored by request to compare to "[2] Mu et al. (2022). Improving Intrinsic Exploration with Language Abstractions. NeurIPS 2022. https://doi.org/10.48550/arXiv.2202.08938" which is probably the most related method and doesn't require costly LLM inference or offline annotation.
>
> C & B) Thank you for providing additional comparisons with E3B. **Please compare E3B, ProgressCounts, [2] (see previous comment), LEMAE all on the same set of benchmarks so that we can see an apples-to-apples comparison.**
>
> D) Thank you for providing additional results in the Appendix for non-symbolic environments. Again, **LEMAE performs worse than the baseline IPPO-DR, questioning the validity of the proposed approach**. At this point I don't suspect LEMAE to work here, but may I suggest an additional visual domain: Vizdoom which has been used in the E3B paper.
>
> As it stands, despite the authors efforts, **none of my concerns have been addressed and I maintain and, based on the negative results in A) and D) compared to baselines, even solidify my assessment of a strong reject**.

---

> > ### Author Response · Authors · 2024-11-19
> > **Severe Misconceptions on Application Scope and Contribution**
> >
> > We thank Reviewer 5ctA's active engagement in discussion. It seems severe misunderstandings about LEMAE still remain. The following continues the rebuttal on contribution, application scope, and experimental setups.
> >
> > **1. Fully Sparse-Reward Version of SMAC and SMACv2: Challenging and Suitable Testbed for Exploration(A)**
> > - (1) We use **fully sparse-reward version of SMAC and SMACv2**, where a non-zero reward is given only at an episode's end to indicate success. This is a **challenging setting in exploration research, as many multi-agent exploration methods fail to even explore a successful state**. In sparse-reward SMACv2, only LEMAE achieves consistent non-trivial performance.
> > - (2) **QMIX-DR refers to QMIX trained with manually designed dense rewards, which is a 'cheating' setup and serves as the performance upper bound**. The performance saturation and success of open-loop policies, which fail in harder maps(Fig.8 of [1]),  that **you mentioned, rely on these dense rewards. It is impressive that LEMAE with sparse rewards matches the performance of QMIX-DR.**
> > - (3) **Several latest MARL works[2,3,4] published in NeurIPS/ICML/AAAI 2024 still use SMAC as a benchmark**. Importantly, we have attached SMAC-v2 results in the initial submission but failed to caught Reviewer 5ctA's attention.
> >
> > **2. [5] is not closely related to LEMAE, and its improved variant has been evaluated(B)**
> > - (1) L-AMIGo and L-NovelD[5] **do not integrate LLMs and require predefined language goals and states, which do not align with our setup**.
> > - (2) Regarding LLM integration: Taking L-AMIGo, which is empirically superior to L-NovelD(Fig.4 in [5]), as an example, **replacing its teacher policy with LLM and removing reliance on predefined language goals result in ELLM**[6]. **ELLM is a more recent method** that uses LLM to generate exploration goals, making it **an improved variant of [5]**. **We have shown LEMAE's superiority by comparing it to ELLM(Fig.3,5)**
> >
> > **3. Methods are evaluated on the same set(C)**
> >
> > We clarify that **LEMAE, E3B, ProgressCounts, and other baselines like ELLM, are evaluated on the same set of benchmark, as shown in Fig.17**. E3B and ProgressCounts are single-agent methods, and ProgressCounts was published after LEMAE's submission. Thus, **they are not directly comparable baselines due to setups and application scopes**. We **evaluate these methods on all MPE tasks in Fig.17 to show LEMAE's consistent superiority.**
> >
> > **4. DR as a 'Cheating' Setup and Disagreement on Repeatedly Unrelated Benchmark Suggestions(D)**
> > - (1) **IPPO-DR refers to IPPO trained with manually designed dense rewards, which is a 'cheating' setup and serves as the performance upper bound.** LEMAE's comparable performance with sparse rewards adequately validates its effectiveness in image-based tasks.  Additionally, LEMAE outperforms E3B on the Visual-Pass task(Fig.12), a single-agent image-based scenario.
> > - (2) We **didn't overstress strong empirical improvements on image-based tasks**. Instead, this work **clearly clarifies the primary application scope(Line 301-305) as efficient multi-agent exploration with symbolic states**, a common setup in prior methods [7]. We **have discussed our innovative designs that address the unique challenges of efficient multi-agent exploration** as shown in previous response. While not the primary focus, we **have demonstrated LEMAE's extension to single-agent and image-based tasks**. Given the sufficient experiments in our manuscript, we believe **it is unreasonable to repeatedly introduce irrelevant benchmarks during the rebuttal process**.
> >
> > **5. Disagree with Overall Evaluation**
> >
> > We appreciate your feedback, but we respectfully disagree with your rating. Our discussions have **focused on the selection of baselines and the boundaries of LEMAE's application scope, without raising concerns regarding the method**. Therefore, we strongly disagree with the current rating, including the **1-score** assessments of soundness, presentation, and contribution, which **differ significantly from those of the other reviewers**. LEMAE is a pioneering attempt to tackle the unique challenges of MARL with LLM for efficient exploration. We kindly request a reconsideration of our submission.
> >
> > ---
> >
> > **Finally, we respect the reviewer's efforts and welcome further discussions on any aspect of LEMAE.**
> >
> > [1] SMACv2: An Improved Benchmark for Cooperative Multi-Agent Reinforcement Learning\
> > [2] Learning Distinguishable Trajectory Representation with Contrastive Loss.NeurIPS 2024\
> > [3] Decoding Global Preferences: Temporal and Cooperative Dependency Modeling in Multi-Agent Preference-Based Reinforcement Learning.AAAI 2024\
> > [4] Individual Contributions as Intrinsic Exploration Scaffolds for Multi-agent Reinforcement Learning.ICML 2024\
> > [5] Improving Intrinsic Exploration with Language Abstractions\
> > [6] Guiding pretraining in reinforcement learning with large language models\
> > [7] Automated Rewards via LLM-Generated Progress Functions

---

> > > ### Comment · Reviewer_5ctA · 2024-11-25
> > >
> > > Thank you for **your** active engagement.
> > >
> > > 1.1 & 1.3) The problems of SMACv1 aren't related to sparse or dense reward settings. As explained in https://arxiv.org/abs/2212.07489, the problem is that in SMACv1 open-loop policies, which observe no ally or enemy information, and can only learn a distribution of actions at each timestep, still do very well. That's why that benchmark is broken and shouldn't be used going forward. This has been stated by SMACv2 authors, many of them authored the original SMACv1 paper. The fact that recent MARL work still publishes on SMACv1 does not provide a valid argument against that. Please show us results on the full SMACv2 benchmark and crucially, make sure it's a fair apple to apple comparison. That is, evaluate baselines (such as QMIX-DR) with the same reward function. Also, please add baselines from other parts of the paper (E3B, ProgressCounts).
> > >
> > > 1.2) Thank you for the clarification. As above, please provide an apple-to-apples comparison where QMIX is actually evaluated on the same MDP. As a meta-point, my "severe misconception" stems from the fact that you plotted the runs of two completely different MDPs (dense vs sparse reward setting) in the same figure.
> > >
> > > 2.) Thank you, I take the argument regarding ELLM already encompassing a comparison to [5]. However, my point from my original review still stands: "In fact, I would go as far and say that the proposed method is not MARL-specific at all, so the authors really need to show that it provides benefits over existing exploration methods in the single agent setting." ELLM has been benchmarked on single-agent environments, in particular Crafter and Housekeep. From my point of view, LEMAE isn't MARL-specific. If you want to claim "LEMAE's superiority by comparing it to ELLM", then demonstrate that on the single-agent environments that ELLM has been using. This goes back to my fair apples-to-apples comparison comment above.
> > >
> > > 3) As per 1), please show results on SMACv2.
> > >
> > > 4.1) Thanks, I don't appreciate the gaslighting here. You, not me, decided to put this "cheating baseline" (arguably a completely different MDP) into the same plot, leading to confusion of the reader (in this case me).
> > >
> > > 4.2) I need to push back in the strongest possible way that the authors are trying to suggest I am "repeatedly introduce irrelevant benchmarks during the rebuttal process". SMACv2 is highly relevant (see point 1). I proposed Vizdoom (that has been used in the E3B paper) as an additional chance for you to provide empirical evidence that your approach works in a non-symbolic domain and thus strengthen the contributions of your paper.
> > >
> > >
> > > Due to the lack of apples-to-apples comparison on a comprehensive set of the SMACv2 benchmark and a fair comparison to ELLM on the singe-agent environments used in their paper, I respectfully disagree with your assessment that LEMAE is a pioneering approach. I hope you will find the feedback above constructive to strengthen your paper in the future.

---

> ### Author Response · Authors · 2024-11-25
> **Thanks for the effort and welcome further discussion**
>
> We thank Reviewer 5ctA's further discussion and post the rebuttal as follows.
>
> 1. Regarding the **evaluation benchmark**, we do not think your viewpoint consistently stands, given several of the latest existing works using SMAC-v1. **We believe the RL community is inclusive enough unless severe bugs happen; otherwise, all recent publications with SMAC-v1 will be problematic according to your criteria.**
> We have clarified that the success of open-loop policies heavily relies on manually designed rewards. Without such rewards, **traditional methods, even those with sufficient information (e.g., QMIX, as shown in Fig. 5), struggle in environments with fully sparse rewards**. This observation reveals the **current performance gap and challenges your characterization of the benchmark as 'broken.'**
> LEMAE achieves strong performance, comparable to results obtained with manually designed dense rewards, which convincingly demonstrates its effectiveness and contribution.
>
> 2. **All setups are apple-to-apple, as we don't modify the MDP transition dynamics**. Some baseline agents can access manually designed **dense rewards, serving as an upper bound**. This setup is appropriate for **evaluating LEMAE's potential** without relying on human-designed reward structures. Recognizing the **empirical limitations of traditional methods like QMIX in fully sparse-reward SMAC**, we exclude them and instead focus on demonstrating LEMAE's universal applicability and its ability to reduce manual effort in reward design, in the supplementary SMACv2 experiments. This is demonstrated through **comparable performance to QMIX-DR(QMIX trained with dense rewards)**. However, to meet your request, we will evaluate ELLM, E3B, and ProgressCount on SMACv2 and report the results in the updated manuscript.
>
> 3. We have clarified previously that **LEMAE is specifically designed to tackle the unique challenges of MARL with LLM for efficient exploration**. These challenges include handling partial observability in MARL, enhancing multi-agent collaboration, alleviating the need for intricate manual reward design, and reducing the complexity of exponentially expanded state spaces. **While the methodology of LEMAE can also be applied to single-agent scenarios, such as utilizing LLM-based discrimination, this is not the primary focus of our application or contributions.**
>
> 4. In the manuscript and previous rebuttal, **we've clearly clarified the scope of LEMAE in symbolic cases**. We have **demonstrated the potential of LEMAE in extending to image-based tasks**, as shown in Appendix F.2. As for pursuing outstanding performance in non-symbolic cases, **we admit it is a meaningful exploration; however, it is not the primary focus of this work and is difficult to investigate within the limited time available for the rebuttal.**
>
> Meanwhile, **it would be appreciated if Reviewer 5ctA adjusted the separate rating, such as presentation, soundness, or contribution. It is kind of a favor to ensure objective and unbiased evaluation** in the RL community. Thanks for your efforts.

---

> > ### Comment · Reviewer_5ctA · 2024-11-25
> >
> > 1)
> > "all recent publications with SMAC-v1 will be problematic according to your criteria" — exactly!
> > "current performance gap and challenges your characterization of the benchmark as 'broken'" — that's not only my assessment but the assessment of two SMACv1 and SMACv2 authors.
> >
> > 2) "we will evaluate ELLM, E3B, and ProgressCount on SMACv2 and report the results in the updated manuscript" — thank you, I am looking forward to that!
> >
> > 3) Then don't claim LEMAE's "superiority" over ELLM. If you want to claim that (since it might actually be true), then compare apples-to-apples on their single-agent RL benchmarks. Also, I still fail to understand what's specific of LEMAE's approach to MARL. As reviewer **Kmbo** pointed out as well "LEMAE is proposed for multi-agent RL problem, but it lacks a consideration for solve MARL training, e.g., cooperation between partners."
> >
> >
> > I don't follow your argument regarding increasing scores on presentation, soundness, or contribution as some favor to ensure objective and unbiased evaluation, and don't appreciate what you might be trying to imply with it.

---

> > > ### Comment · Reviewer_5ctA · 2024-11-26
> > >
> > > Further on your point of "**All setups are apple-to-apple, as we don't modify the MDP transition dynamics**": an MDP is literally defined as 4-tuple <S, A, P, R>, so the moment you change the reward function R, you have a different MDP. Thus it shouldn't be surprising that plotting results from different MDPs into the same figure can cause confusion.
> > >
> > > Hope this helps.

---

> ### Author Response · Authors · 2024-11-26
> **Comprehensive Explanations to Avoid Misunderstandings**
>
> **Thank you for your active engagement. The following clarifies any potential misunderstandings.**
>
> 1. We wish to clarify that it is **the empirical results that highlight the significant performance gap in fully sparse-reward SMAC**.
> In scientific research, we believe that **conclusions should be drawn from empirical evidence, and that specific settings must be carefully analyzed, rather than relying on prior perspectives**.
> Your point regarding the benchmark being 'broken' seems to stem from its ease in dense-reward scenarios. However, we have demonstrated that **sparse-reward SMAC presents substantial challenges**, which were not the settings discussed by the SMAC authors. Therefore, we believe the assertion that it is 'broken' may be not justified.
> 2. **Additional baseline results on SMACv2 have been included in the updated manuscript** as a supplement to demonstrate **LEMAE's effectiveness in incorporating LLM for efficient multi-agent exploration, compared to previous LLM-based single-agent methods**.
> 3. **We do not overstate the 'superiority' of our method**. In the manuscript, we objectively compare the performance of ELLM and LEMAE in multi-agent scenarios (the primary focus of this work), as shown in Fig. 3 and 5. We have made appropriate adaptations to ELLM for multi-agent scenarios to **ensure a fair, apples-to-apples comparison**.
> 4. We clarify that **our comparisons are 'apples-to-apples'**. The underlying MDP (excluding intrinsic reward) is the same across all methods, with a sparse reward function. **Methods using dense rewards can be viewed as sparse-reward MDP + manually designed intrinsic rewards**. We emphasized the unmodified transitions in the previous response to prevent any confusion. **Introducing a human expert as an upper-bound baseline is a standard practice[1,2] to demonstrate the potential of the methods.** Besides, the evaluation metric is test win rate, an objective and fair metric
> for all methods. Thank you for your suggestions. To avoid any confusion, we have provided additional clarification in the revised manuscript.
> 5. We have previously clarified that LEMAE is specifically designed to **address the unique challenges of MARL** with LLM for efficient exploration. Regarding multi-agent cooperation, we have clarified before that, as shown in Sec. 4.2, we explicitly prompt LLM to **prioritize role division among agents to enhance multi-agent collaboration**. This straightforward design yields impressive results: the visitation map in Fig.1(d) shows an organic division of labor.
> We appreciate your recognition of LEMAE's potential in single-agent scenarios. However, this is **not the primary focus of this work**, as stated in 4. We intend to further **advance the core idea behind LEMAE in future research, i.e., leveraging LLM-based discrimination to incorporate knowledge into efficient RL algorithms.**
>
> **We appreciate your kindness and efforts to improve the quality of our work. We hope our response provides a clearer and more comprehensive basis for evaluating our work without misunderstandings.**
>
> [1] Lazy Agents: A New Perspective on Solving Sparse Reward Problem in Multi-agent Reinforcement Learning. ICML 2023\
> [2] Eureka: Human-level reward design via coding large language models. ICLR 2024

---

### Official Review · Reviewer_AW9u · 2024-10-28

**Soundness:** 3
**Presentation:** 2
**Contribution:** 3
**Rating:** 6
**Confidence:** 4

**Summary:**

This paper proposes to channel informative task-relevant guidance from a knowledgeable LLM for efficient Multi-Agent Exploration. It uses LLM to localize key states and use hindsight reward for exploration (also use memory tree to orginaze the key states). The results show that the proposed methods achieve better results than most "traditional" deep MARL methods.

**Strengths:**

1. important topic and novel methods: Efficient multi-agent exploration is important for MAS, the proposed methods leverage LLM for this goal, which is novel as the reviewer knows.

2. memory tree to orginaze the key states for guided exploration (instead of greedy redundant exploration) is interesting and novel.

3. The paper is clear and easy to understand.

4. The results show that the proposed methods achieve better results than most baseline methods.

**Weaknesses:**

1. Proposition 4.1 is one of hte most important motivation for task-relevant information based exploration. But the setting is the one-dimensional asymmetric random walk problem. What is means for a more realistic scenarios (e.g., MPE, SMAC)?  Could the authors discuss how the insights from Proposition 4.1 might generalize or relate to more complex environments (higher-dimensional state spaces with multiple agents) like MPE and SMAC?

2. The key states localization is the most important for the proposed methods. But current description (i..e, Section 4.2)  is not clear about the method detaills. (besides discrimination by LLM rather than generation; m discriminator functions). Could the authors discuss the process of generating the m discriminator functions (or the detailed prompts), how the number m is determined, or how the discriminator functions are applied to states in practice. And how to make sure the LLM-based discriminators can work well as the designed intention?

3. The baselines compared seems to be general MARL methods. It is better to compare MA-exploration methods and methods that also combine LLM with MARL (e.g., [1,2,3]).  Could the authors explain why these specific baselines were not included, and could you either add comparisons to these methods or discuss how their approach differs conceptually from these related works. This would help contextualize your contribution within the most relevant recent literature.

[1] Learning When to Explore in Multi-Agent Reinforcement Learning. TCYB 2023.  (which learns to when to explore)
[2] Controlling Large Language Model-based Agents for Large-Scale Decision-Making: An Actor-Critic Approach. (which uses two LLMs for exploration-exploitation tradeoff)

**Questions:**

see weakness.

---

> ### Author Response · Authors · 2024-11-18
>
> _We sincerely appreciate Reviewer AW9u's efforts and positive feedback on the importance, novelty, presentation, and impressive experiments. The following addresses the concerns._
>
> **1.Motivation Discussion**
>
> Proposition 4.1 provides a quantitative analysis of the efficacy of incorporating key states.
> **For more realistic tasks, we illustrate using the *Pass* task** as follows:\
> As illustrated in Fig.1(c), without key states (CMAE), agents must explore the entire room randomly from their initial positions until they reach a success state. This results in significant redundant exploration.\
> In contrast, **with key states incorporated in LEMAE, exploration becomes more organized and targeted**: Agents first focus on finding key state $\kappa_1$. Once $\kappa_1$ is located, agents can skip exploring areas preceding $\kappa_1$ and focus on states that follow it.\
> Both the visitation map in Fig.1(d) and the performance in Fig.3 illustrate that **key states provide a significant advantage through reducing redundant exploration.**\
> Moreover, in our MARL design, **key states encourage role specialization among agents to enhance multi-agent collaboration**: The visitation map in Fig.1(d) reveals an emergent division of labor, while the discrimination functions in Fig.4 provide an intuitive explanation of this phenomenon.
>
> **2.Implementation Details Explanation**
>
> We apologize for any confusion and **have revised the manuscript for improved clarity**. Below is an explanation of the implementation details for key state localization:
>
> - Prompts and Generation: Due to page limits, **detailed prompts and responses are in Appendix D**. Specifically, as shown in **Sec.4.3**, we design a prompt template with role instructions to help LLM understand tasks and define key states, along with output constraints for responding in a specific JSON format. For each new task, LLM takes in the prompt template with task details (task description and state form) and generates key states definitions and discriminator functions. **The discriminator functions can be easily extracted from the JSON response via key-value matching.**
>
> - Key States Number $m$: We **do not strictly constrain $m$, allowing LLM to determine it** while using prompts to discourage overly low values. Specific details and values are in **Appendix D**.
>
> - Applied to states: Each discriminator function $\mathcal{F}_i$ takes the state $s_t$ at timestep $t$ as input and outputs a boolean value to tell whether $s_t$ is the corresponding key state $\kappa_i$(**Sec.4.3**).
>
> - Response Quality Assurance: As detailed in **Sec.4.3**, we introduce a **Self-Check mechanism to ensure LLM response quality**. It involves two steps: LLM rethinking for self-assessment with a set of queries and code verification to test discriminator function executability on actual inputs, iterating until successful. **Table 1 shows the proposed mechanism's effectiveness and high LLM response quality**.
>
> **3.Baselines Clarification**
>
> We want to clarify that **all the MARL baselines we adopted, except for IPPO and QMIX, are widely recognized multi-agent exploration methods** in prior works[1,2]. As suggested, we have **added discussions on WToE [2] and LLaMAC [3] in Sec. 3 and evaluated WToE in Appendix F.8**. Below, we provide additional discussions and explain the rationale for not evaluating LLaMAC:
>
> > A recent influential work, WToE, focuses on when to explore by identifying discrepancies between the actor policy and a short-term inferred policy that adapts to environmental changes, which does not employ intrinsic rewards for guidance. LLaMAC employs multiple LLMs to balance exploration and exploitation, emphasizing step-wise decision-making via frequent LLM calls with rewards as feedback rather than RL-based low-level exploration with sparse rewards(our setup).
>
> [1]Lazy agents: a new perspective on solving sparse reward problem in multi-agent reinforcement learning.\
> [2]Learning When to Explore in Multi-Agent Reinforcement Learning. TCYB 2023.\
> [3]Controlling Large Language Model-based Agents for Large-Scale Decision-Making: An Actor-Critic Approach.

---

> > ### Author Response · Authors · 2024-11-25
> > **Looking forward to Continuing the Discussion**
> >
> > **Dear Reviewer AW9u**:
> >
> > We sincerely thank you for your efforts and valuable feedback on reviewing our paper. We noticed that you have not yet participated in the discussion, and we would like to ask if you have any additional comments or questions that we can address collaboratively. **We have revised the manuscript based on your insightful feedback and are eager to address any remaining concerns.**
> >
> > Please let us know if you have any further thoughts or questions. We look forward to continuing our discussion.
> >
> > Best regards,
> >
> > Authors

---

> > > ### Comment · Reviewer_AW9u · 2024-11-26
> > > **I support the acceptance of this paper; I will keep my score**
> > >
> > > I read the response, which addresses most of my concerns. However, Proposition 4.1 for realistic scenario is not clear. Besides, I think the key state localization method is rather limited to specific scenarios. Nevertheless, the overall idea is cool and the authors prove this works for some environments, and I support the acceptance of this paper (but did not change my score)

---

> > > > ### Author Response · Authors · 2024-11-26
> > > >
> > > > We sincerely thank Reviewer AW9u for the recognition of our innovation, as well as the positive assessment of its soundness and contribution. This work represents a first attempt to achieve efficient multi-agent exploration using LLM priors and is primarily focused on certain scenarios. LEMAE also has the potential for extension to other scenarios in the future, such as image-based tasks in Appendix F.2.
> > > >
> > > > We will further polish the manuscript in accordance with your valuable suggestions.
> > > > All of the suggested experiments and analyses have been included in the updated manuscript.
> > > > Many thanks.

---

### Official Review · Reviewer_kuvr · 2024-11-04

**Soundness:** 3
**Presentation:** 3
**Contribution:** 2
**Rating:** 5
**Confidence:** 4

**Summary:**

This paper proposes to improve exploration in multi-agent environments by generating key states with the aid of LLMs and then use hindsight experience replay to generate auxiliary rewards. Specifically, task descriptions as well as task-relevant knowledge is converted to text and then fed to an LLM that is tasked to generate a function (e.g., written in Python) that takes in a state and returns whether this is a goal state. During a typical online reinforcement learning procedure, each experienced state is examined by the key state checking function, and if it is masked as a key state, hindsight experience replay will be used to add auxiliary rewards to prior states in the same trajectory/episode. The proposed algorithm was tested on the Multiple-Particle Environment and the StarCraft Multi-Agent Challenge.

**Strengths:**

Hindsight experience replay is a very effective auxiliary reward generation methods that have shown successes in many RL environments. However, its main limitation is the difficulty to correctly “identify” goal states. This paper proposes to use LLM to generate such “key states”.

**Weaknesses:**

- Connection with other LLM-based reward shaping algorithms. While the paper discusses many LLM-based reward generation algorithms in their related works, better and more comprehensive comparison with these methods are needed in the experiment sections to better demonstrate the effectiveness of the proposed algorithm (e.g., Xu et al., 2023 and Liu et al., 2023 cited in the paper). Specifically, as mentioned in the paper, specific prompts for description the task/state space/action space are used for each environment independently. Whether the same set of prompts are used in relevant baselines? If the prompt is good enough for the LLM to generate key state description functions, it seems natural that with similar prompts the LLM can also directly generate good auxiliary rewards and/or reward functions, which are done by many existing LLM-based reward generation algorithms. It would be very helpful to clearly describe the settings and make sure that the relevant baselines are also exposed to the same prompts.

- There is only one LLM-based reward generation baseline adopted. Are there particular reasons for only including this LLM-based baseline algorithm? Since the proposed method leverages knowledge from LLMs, it would be more natural to include more LLM-based baselines.

- Generalizability of the LLM-based key-state detector to other environments. The adopted environments have discrete and small state/action spaces, which makes it easier to create key-state detectors. Also, the environments are relatively simple to describe with natural language. I wonder how this method performs in environments with more complex state space/action space/transition dynamics, for example in robotics tasks?

- Minor: relation with Go-Explore [1]. This key states memory tree seems to be related to go-explore. Could the authors discuss their similarities and differences?


[1] Ecoffet, Adrien, Joost Huizinga, Joel Lehman, Kenneth O. Stanley, and Jeff Clune. "Go-explore: a new approach for hard-exploration problems." arXiv preprint arXiv:1901.10995 (2019).

**Questions:**

- How does the proposed method compare with other LLM-based reward shaping algorithms?
- Could the authors clarify whether the same prompts were used across all LLM-based methods?
- How well can the proposed algorithm scale to environments with complex state space/action space/transition dynamics?
- It seems that the proposed method can be applied to both single-agent and multi-agent environments. Is there a particular reason to use it primarily in multi-agent environments?
- How does the key states memory tree compare to the memory structure used in Go-Explore?

---

> ### Author Response · Authors · 2024-11-18
>
> _We sincerely appreciate Reviewer kuvr's efforts and some positive feedback. The following addresses the concerns._
>
> **1.Connection with LLM-based Methods**
>
> - Connections:\
> (1)**We evaluated the most relevant LLM-based methods in Fig.3**: LLM step-wise exploration goals generation, exemplified by ELLM[1], and LLM-based reward design, exemplified by Eureka[2].
> **LEMAE utilizes LLM's discrimination and coding capabilities, eliminating the need for state captioner and frequent LLM calls** in ELLM, **as well as the high information demands**(e.g., environment codes in Eureka) and **reliability issues**(as discussed in Appendix B.1) associated with reward design.\
> (2)**As suggested, we evaluate ProgressCounts[3], a recent LLM-based method, in revised Appendix F.8, which combines LLM reward design and count-based exploration**. LEMAE consistently outperforms it, **highlighting the effectiveness of LEMAE in better bridging LLM with MARL**
> |task|LEMAE|ProgressCounts|
> |-|-|-|
> |Pass|**1.0±0.0**|0.20±0.40|
> |Push-Box|**1.0±0.0**|0.71±0.41|
>
> - Fair Information: We keep **consistent prompt information across all LLM-based methods to ensure fairness**, including task descriptions and state forms. This has been clarified in the manuscript.
>
> - Reward Generation Failure:
> **In Appendix B.1, we discussed the insights behind LLM discrimination and the challenges of direct reward generation**: Discrimination is preferable to generation as it requires less task information requirements and allows for easier error detection and correction. Intuitively, in **partially observable** tasks like Pass, key state/reward generation is unreliable and even infeasible due to missing information, such as hidden switch positions, **explaining Eureka's poor performance**.
>
> **2.Applications in Robotics Control**
>
> As suggested, we **evaluate LEMAE on MaMuJoCo[4], a MARL robotics benchmark, in Appendix F.7 of the revised manuscript**. We adapt the tasks to emphasize exploration with sparse rewards for achieving high velocity.
>
> |task|LEMAE|HAPPO(dense reward)|HAPPO|
> |-|-|-|-|
> |HalfCheetah|1.00±0.00|1.00±0.00|0.00±0.00|
> |Humanoid|0.80±0.40|0.94±0.13|0.00±0.00|
>
> LEMAE is comparable to the baseline trained with human-designed dense rewards. **This aligns with previous conclusions and indicates its potential for complex tasks, owing to the reliability of LLM key state discrimination.** Notably, **the extensions to single-agent RL(Appendix F.4), image-based tasks(Appendix F.2) and robotics applications(here) demonstrate LEMAE is a general method for bridging LLM and RL for efficient exploration, with potential applicability to diverse and complex tasks.**
>
> **3.Explanation for MA**
>
> We clarify that this work represents **a pioneering attempt to address efficient exploration in MARL with LLM, an important domain with broad applications and a current research gap**. Compared to prior LLM-based methods that primarily target single-agent scenarios, LEMAE is designed to **tackle the unique challenges in MARL**: (1) LLM-based key states discrimination to provide reliable priors during training, particularly for **high-dimensional and partially observable MARL tasks**; (2) Labor-division-oriented prompt designs to **reduce manual effort and foster multi-agent cooperation**; (3) Subspace-based Hindsight Intrisic Reward to **improve policy guidance and alleviate manual reward design**; and (4) Key States Memory Tree to organize exploration with memory, **reducing complexity by focusing on a meaningful state subspace within the exponentially expanded state space**.
>
> Despite primarily focusing on multi-agent tasks, **LEMAE is a general method for improving efficient exploration in RL** and can also be easily **extended to single-agent settings**, as shown in Appendix F.4.
>
> **4.Connections with Go-Explore**
>
> Thanks for the suggestion. Go-Explore[5] is an influential work tackling exploration in RL.
> - The **similarities** between our KSMT and the archive in Go-Explore lie in both methods organizing exploration through memory, i.e., by selecting possible historical states to explore.\
> - The **differences and partial contributions of LEMAE** are as follows: (1) Key states are **semantically meaningful and task-critical**, whereas the archived states in Go-Explore are randomly explored; (2) Our KSMT samples key states based on **actual key states transitions**, enhancing its reliability; (3) We propose *Explore with KSMT* to balance exploration and exploitation, thereby **reducing exploration complexity by focusing on a more meaningful state subspace**.
>
> We've cited Go-Explore and added the above discussion in updated manuscript Appendix F.5.1.
>
> [1]Guiding pretraining in reinforcement learning with large language models\
> [2]Eureka: Human-level reward design via coding large language models\
> [3]Automated Rewards via LLM-Generated Progress Functions\
> [4]Facmac: Factored multi-agent centralised policy gradients\
> [5]Go-explore: a new approach for hard-exploration problems

---

> > ### Author Response · Authors · 2024-11-25
> > **Looking forward to Continuing the Discussion**
> >
> > **Dear Reviewer kuvr**:
> >
> > We sincerely thank you for your efforts and valuable feedback on reviewing our paper. We noticed that you have not yet participated in the discussion, and we would like to ask if you have any additional comments or questions that we can address collaboratively. **We have revised the manuscript based on your insightful feedback and are eager to address any remaining concerns.**
> >
> > Please let us know if you have any further thoughts or questions. We look forward to continuing our discussion.
> >
> > Best regards,
> >
> > Authors

---

> ### Author Response · Authors · 2024-11-29
>
> **Dear Reviewer kuvr**,
>
> We greatly appreciate your time and effort in reviewing our paper. We want to check if responses have addressed your concerns as the deadline is approaching.
>
> **Your engagements mean a lot to us, and it would be appreciated if you reconsider the evaluation after reading the rebuttal and the updated manuscript**. Importantly, thanks for your constructive suggestions.
>
> Best regards,
>
> Authors

---

### Official Review · Reviewer_Kmbo · 2024-11-10

**Soundness:** 3
**Presentation:** 3
**Contribution:** 2
**Rating:** 6
**Confidence:** 3

**Summary:**

This paper study utilizing LLMs to improve exploration for multi-agent RL setting. The key motivation is that LLMs, pre-trained on general knowledge, can extract task-relevant guidance for improving exploration. The proposed method,  LEMAE, identifies key states by prompting LLMs, and encourages the agents to move forward key states using a memory tree. Experiment results on SMAC and MPE show that LEMAE outperforms various RL methods in terms of improvement speed and final performance.

**Strengths:**

Developing an effective way to leverage the linguistic knowledge in LLMs for RL is a promising approach. This paper makes a step forward by proposing the idea of 'discriminating key state' and tree search. I believe this could be a valuable supplement to RL community with the integration of large foundation models.

In addition, I list some other pros as follow:

1. Overall the paper is well-organized and well-established. It is easy to follow the motivation and method design.
2. The proposed method seems a general method that can be utilized for wide RL problems, e.g., single agent setting. It would be interesting to see the results on single agent RL tasks.
3. The experiment is comprehensive, including wide ranges of baselines and evaluation tasks.

**Weaknesses:**

It seems that LEMAE is quite limited to specific tasks, whose state is simple to describe. A key point of LEMAE is to address the symbolic state representation issue for LLMs. However, for tasks with complex state representation, such as robotics control, it is hard to explain a specific state value. LEMAE is hard to be applied in such domain, which is an important sub-filed of RL tasks.

Besides, LEMAE is proposed for multi-agent RL problem, but it lacks a consideration for solve MARL training, e.g., cooperation between partners. Taking this into consideration would make the method more reliable and efficient.

There are also some more suggestions on the writing:
1. I feel that the introduction of the method in Section 1 is a simple re-writing of the abstract.
2. I do not find the importance of putting "CHOICES ARE MORE IMPORTANT THAN EFFORTS" in the title. I think it is not the main highlight of this paper. Maybe it is better to consider from the perspective of LLMs.

**Questions:**

Refer to weakness.

---

> ### Author Response · Authors · 2024-11-18
>
> _We sincerely appreciate Reviewer Kmbo's efforts and positive feedback on the importance, broad application, and comprehensive experiments. The following addresses the concerns._
>
> **1.Applications in Robotics Control**
>
> Following your suggestion, we **evaluate LEMAE on MaMuJoCo [1], a MARL robotics benchmark**. We adapt the tasks to emphasize exploration with sparse rewards for achieving high velocity.
>
> |task|LEMAE|HAPPO(dense reward)|HAPPO|
> |-|-|-|-|
> |HalfCheetah|1.00±0.00|1.00±0.00|0.00±0.00|
> |Humanoid|0.80±0.40|0.94±0.13|0.00±0.00|
>
> LEMAE achieves performance comparable to the baseline trained with human-designed dense rewards. This observation is consistent with previous conclusions. **LEMAE benefits from the reliability of the proposed LLM key states discrimination, and the results underscore LEMAE’s potential for handling complex tasks**. Additional details have been added into the revised manuscript(**Appendix F.7**).
>
> In fact, **the use of discrimination instead of direct generation has relaxed the application assumption** to only state semantics(which can be further eliminated with multi-modal models), eliminating the need for environment codes or state captioners required by prior methods, as discussed in Appendix B.
> **LEMAE has been extended to single-agent RL(Appendix F.4), image-based tasks(Appendix F.2) and robotics applications(here), implying that it is a general method for bridging LLM and RL for efficient exploration, with potential applicability to diverse and complex tasks.**
>
> **2.Clarification of MARL Consideration**
>
> We clarify that this work represents **a pioneering attempt to address efficient exploration in MARL with LLM, an important domain with broad applications and a current research gap**. Compared to prior LLM-based methods that primarily target single-agent scenarios, LEMAE is designed to **tackle the unique challenges in MARL**: (1) LLM-based key states discrimination to provide reliable priors during training, particularly for **high-dimensional and partially observable MARL tasks**; (2) prompt designs to **reduce manual effort and foster multi-agent cooperation through encouraging labor division among agents**; (3) Subspace-based Hindsight Intrisic Reward to **improve policy guidance and alleviate manual reward design**; and (4) Key States Memory Tree to organize exploration with memory, **reducing complexity by focusing on a meaningful state subspace within the exponentially expanded state space**.
>
> Specifically, regarding (2), as shown in Sec. 4.2, we explicitly prompt LLM to **prioritize role division among agents to enhance multi-agent collaboration**. This straightforward design yields impressive results: Fig. 1(d)'s visitation map shows **an organic division of labor emerging**, while the discrimination functions in Fig. 4 provide an intuitive explanation of this formation.
>
> **3.Writing improvement**
>
> Thanks for your suggestions. We have polished the method introduction in Sec.1.\
> Regarding the phrase *Choices are More Important than Efforts* in the title, we used it to emphasize our idea that choosing task-relevant guidance is better for exploration efficiency in RL than expending aimless or redundant effort. LLM is chosen as a tool to provide this guidance, as it has been shown to possess priors for task understanding across various tasks.
>
> **References:**
>
> [1] Facmac: Factored multi-agent centralised policy gradients

---

> > ### Author Response · Authors · 2024-11-25
> > **Looking forward to Continuing the Discussion**
> >
> > **Dear Reviewer Kmbo**:
> >
> > We sincerely thank you for your efforts and valuable feedback on reviewing our paper. We noticed that you have not yet participated in the discussion, and we would like to ask if you have any additional comments or questions that we can address collaboratively. **We have revised the manuscript based on your insightful feedback and are eager to address any remaining concerns.**
> >
> > Please let us know if you have any further thoughts or questions. We look forward to continuing our discussion.
> >
> > Best regards,
> >
> > Authors

---

> > > ### Comment · Reviewer_Kmbo · 2024-11-26
> > >
> > > Thanks for your response. I appreciate your additional experiments on robotics control. I still can not find the necessarily of applying LEMAE on MARL problem. Maybe the proposed techniques are more suitable for single-agent setting due to the complexity of key state search of MARL setting. In general, I acknowledge the author response has addressed parts of my concerns. Thus I tend to maintain my positive assessment.

---

> > > > ### Author Response · Authors · 2024-11-26
> > > >
> > > > We sincerely thank Reviewer Kmbo for positive assessment and helpful suggestions. We'll polish the manuscript and further highlight LEMAE's role in MARL. All your suggested experiments and analysis is included in the updated manuscript. Many thanks.

---

### Author Response · Authors · 2024-11-18
**Global Response**

_We sincerely thank all reviewers and area chairs for their work. This global response summarizes reviews, addresses concerns, answers questions, and reports changes in the manuscript._

---
**I. Review Summary**
-

We thank all reviewers for their positive comments:
- A promising and important topic **[#Reviewer Kmbo, AW9u, 5ctA]**
- A sound, novel and general method **[#Reviewer kuvr, AW9u, Kmbo]**
- A well-established paper easy to follow **[#Reviewer Kmbo, AW9u]**
- Impressive and comprehensive experiments **[#Reviewer Kmbo, AW9u]**

---
**II. Primary Concerns and Questions**
-
**1.Applicability to Robotics Control[#Reviewer kuvr, Kmbo] and Non-Symbolic[#Reviewer 5ctA] Tasks**
- We clarify that **this work primarily focuses on achieving efficient exploration in tasks with symbolic state, a common assumption in prior methods[1,2]**. We innovatively **propose utilizing LLM discrimination with effective intrinsic reward and tree exploration components, rather than direct generation**. This approach relaxes the application assumption and shows impressive effectiveness in MARL.
- As suggested, we **extended LEMAE to MaMuJoCo[3], a MARL robotics benchmark, and *Visual-Pass*, an image-based task**. As shown in **Appendix F.2 and F.7 of the revised manuscript**, the consistent success of LEMAE demonstrates its **generality in bridging LLM and RL for efficient exploration, indicating its potential applicability to diverse and complex tasks**.

**2.Clarification of MARL Consideration[#Reviewer Kmbo, kuvr, 5ctA]**

We clarify that this work represents **a pioneering attempt to address efficient exploration in MARL with LLM, an important domain with broad applications and a current research gap**. Compared to prior LLM-based methods that primarily target single-agent scenarios, LEMAE is designed to **tackle the unique challenges in MARL**: \
(1) LLM-based key states discrimination to provide reliable priors during training, particularly for **high-dimensional and partially observable MARL tasks**; \
(2) Labor-division-oriented prompt designs to **reduce manual effort and foster multi-agent cooperation**; \
(3) Subspace-based Hindsight Intrisic Reward to **improve policy guidance and alleviate manual reward design**; and \
(4) Key States Memory Tree to organize exploration with memory, **reducing complexity by focusing on a meaningful state subspace within the exponentially expanded state space**.

Experimentally, we **evaluate LEMAE on standard multi-agent exploration benchmarks (MPE, SMAC, SMAC-v2)**, showing its superiority through fair comparision with multi-agent exploration baselines and relevant LLM-based methods.

**3.Clarification on Comparison with LLM-based Methods[#Reviewer kuvr, AW9u, 5ctA]**
- **We evaluated the most relevant LLM-based methods in Fig.3**: LLM step-wise exploration goals generation, exemplified by ELLM[4], and LLM-based reward design, exemplified by Eureka[1]. **LEMAE utilizes LLM's discrimination and coding capabilities, eliminating the need for state captioner and frequent LLM calls** in ELLM, **as well as the high information demands**(e.g., environment codes in Eureka) and **reliability issues**(as shown in Fig.3 and discussed in Appendix B.1) associated with reward design.
- **As suggested, we evaluate ProgressCounts[2], a recent LLM-based method, in revised Appendix F.8, which combines LLM reward design and count-based exploration**. LEMAE consistently outperformed it, **highlighting the superior effectiveness of LEMAE in better bridging LLM with MARL.** Some of the suggested baselines are not applicable to our setup, and we have added further discussion in Sec. 3 of the revised manuscript.

**4.Other questions and opensource plan**

Thanks and we leave other questions and answers in individual responses. *We'll opensource codes to advance research on leveraging LLMs' prior knowledge for RL.*

**References:**

[1]Eureka: Human-level reward design via coding large language models\
[2]Automated Rewards via LLM-Generated Progress Functions\
[3]Facmac: Factored multi-agent centralised policy gradients\
[4]Guiding pretraining in reinforcement learning with large language models

---
**III. Manuscript Changes**
-

1.Incorporate new experimental results, including additional baselines and extended tasks.

2.Add descriptions and discussions, address suggested revisions and fix typos.

---
*Once again, thank you to all the reviewers and area chairs. Your effort means a lot in improving our manuscript.*

---

### Author Response · Authors · 2024-11-23
**Looking forward to Continuing the Discussion**

Dear Reviewers,

We hope this message finds you well. Thank you for taking the time to review our paper. We sincerely appreciate your thoughtful feedback and believe that engaging in discussion during the rebuttal period could be mutually beneficial in enhancing the paper’s quality.

As the discussion phase is nearing its conclusion, we would like to kindly ask if you have any additional comments or questions that we could address together. Your expertise and insights would be invaluable in refining the paper, and we are eager to address any remaining concerns.

Please feel free to share any thoughts or questions. We look forward to continuing the discussion.

Best regards,

Authors

---

### Comment · Area_Chair_441r · 2024-11-25

Dear Reviewers,


This is a friendly reminder that the discussion will end on Nov. 26th (anywhere on Earth). If you have not already, please take a close look at all reviews and author responses, and comment on whether your original rating stands.


Thanks,

AC

---

### Author Response · Authors · 2024-11-27
**Manuscript Revisions**

Dear Reviewers,

We hope this message finds you well. We sincerely thank all the reviewers for their time and effort in reviewing our work. In response to the valuable suggestions, we have **updated the manuscript as follows**:
- **Writing improvements**: We have polished the manuscript to correct typos, add discussions, and enhance clarity, addressing potential ambiguities.
- **Expanded applications**: We have demonstrated the broader applicability of LEMAE, including its use in **robotic control (Appendix F.7) and image-based tasks (Appendix F.2)**.
- **Comprehensive baseline comparisons**: To **better illustrate the efficacy of our design choices in integrating LLMs for efficient multi-agent exploration**, apart from two previously evaluated LLM-based methods (ELLM and Eureka), we compare LEMAE with a generic intrinsic reward method (E3B), a very recent LLM-based method (ProgressCount), and a multi-agent exploration method (WToE). The first two were originally designed for single-agent setups. **Results from MPE (Appendix F.8) and SMACv2 (Appendix F.1) highlight the outstanding performance of LEMAE, confirming the effectiveness of our design choices in integrating LLMs for MARL**.

Once again, we would like to express our sincere gratitude to all the reviewers for their constructive feedback and invaluable contributions to improving this work. Your insights have greatly enhanced the quality of this manuscript.

Best regards,

Authors

---

### Comment · Area_Chair_441r · 2024-11-29

Dear Reviewers,

This is a friendly reminder that the last day that reviewers can post a message to the authors is Dec. 2nd (anywhere on Earth). If you have not already, please take a close look at all reviews and author responses, and comment on whether your original rating stands.

Thanks,

AC

---

### Meta-Review · Area_Chair_441r · 2024-12-20

**Metareview:**

This paper proposes to improve exploration in multi-agent environments by generating key states with the aid of LLMs and use hindsight experience replay to generate auxiliary rewards.

The method is novel, the paper is well-organized, and it is easy to follow the motivation and method design.

However, the method is not sufficiently evaluated in more complex tasks and it also lacks consideration of specifics in multi-agent RL. The baselines are not sufficient for evaluating the proposed methods. Thus, current empirical results do not fully support the claim of the paper.

**Additional Comments On Reviewer Discussion:**

During rebuttal, more experiments on robotic controls and SMACv2 and new baselines are added. I think these newly added experiments should be the main results, thus the authors should include more results on these environments and resubmit the paper.

---

### Decision · Program_Chairs · 2025-01-22

Reject